# A preclinical secondary pharmacology resource illuminates target-adverse drug reaction associations of marketed drugs

Jeffrey J. Sutherland[1], Dimitar Yonchev[2], Alexander Fekete[1] & Laszlo Urban [1]✉

In vitro secondary pharmacology assays are an important tool for predicting clinical adverse drug reactions (ADRs) of investigational drugs. We created the Secondary Pharmacology Database (SPD) by testing 1958 drugs using 200 assays to validate target-ADR associations. Compared to public and subscription resources, 95% of all and 36% of active (AC50 < 1 μM) results are unique to SPD, with bias towards higher activity in public resources. Annotating drugs with free maximal plasma concentrations, we find 684 physiologically relevant unpublished off-target activities. Furthermore, 64% of putative ADRs linked to target activity in key literature reviews are not statistically significant in SPD. Systematic analysis of all target-ADR pairs identifies several putative associations supported by publications. Finally, candidate mechanisms for known ADRs are proposed based on SPD off-target activities. Here we present a freely-available resource for benchmarking ADR predictions, explaining phenotypic activity and investigating clinical properties of marketed drugs.

Adverse drug reactions (ADRs) are a significant cause of drug discovery and clinical program terminations and post-marketing drug withdrawals[1]. Further, ADRs are a frequent cause of patient drug discontinuation, increasing disease burden for patients and the healthcare system[2]. Anticipating the ADR profile of investigational drugs during lead optimization allows drug discovery teams to pursue strategies for reducing the safety liability while maintaining favorable on-target pharmacological properties.

ADRs mediated by unintended drug activity may involve interaction with one or more targets in the druggable proteome[3]. Despite advances in high-throughput transcriptomic, proteomic, or cellular imaging techniques for predicting ADRs[4], panels of in vitro biochemical and cellular assays measuring the effect of drugs on key protein targets retain their pre-eminence in preclinical secondary pharmacology testing[5,6]. However, the number of targets with well-established roles in mediating ADRs is limited. Examples include hERG (KCNH2) for QT prolongation, α1A adrenergic receptor (ADRA1A) modulation for arrhythmia (agonists) or orthostatic hypotension (antagonists), and dopamine D1 (DRD1) antagonism for dyskinesia and tremors[7]. Beyond

the hERG channel, a lack of scientific consensus on the strength of the evidence linking target activity to ADRs may contribute to the high variability in panel composition across the pharmaceutical industry[8].

Prior studies have explored relationships between activity results from biochemical in vitro assays and ADRs from marketed drugs[9–12]. These studies have been qualitative in nature (e.g., citing literature implicating the target), were limited to curated activity results from resources such as ChEMBL[13] and DrugCentral[14], and generally used measures of activity potency that did not account for variable human pharmacokinetic properties of drugs, namely the maximal drug exposure (Cmax) at the highest approved dose. Recently, Smit et al.[15] reported the first systematic analysis of safety margin vs. ADR relationships using biochemical activity and human exposure results from ChEMBL, and identified 45 targets with statistically significant relationships vs. human ADRs. Because results from ChEMBL are parsimonious (i.e., most assays vs. compound pairs have no reported results from the literature), the authors used QSAR modeling to fill in missing values and could not account for potential confounding relationships when establishing statistical significance.

[1]Novartis Institutes for Biomedical Research, Cambridge, MA, USA. [2]Novartis Institutes for Biomedical Research, Basel, Switzerland.
✉e-mail: laszlo.urban@novartis.com

Over the course of several years, we have systematically evaluated the activity of 1958 drugs vs. panels of biochemical and cellular in vitro assays to create a secondary pharmacology database (SPD). Unusually for such resources, all compounds were tested at 8 or more concentrations, with the concentration resulting in 50% of maximal activity ($AC_{50}$) available for all tested drug vs. assay pairs. The database reports ca. 150 000 $AC_{50}$ values for marketed drugs, allowing systematic analysis of target (assay) vs. ADRs reported in databases such as SIDER[16] and the FDA adverse drug reaction reporting system (FAERS). To our knowledge, the only comparable resource is the Eurofins BioPrint database[17] which is available by subscription only. Here, we report overall low concordance between results from the SPD (obtained using a limited number of assay protocols for each target) vs. results from ChEMBL and DrugCentral (obtained using a wide variety of such protocols). We illustrate the utility of the database by identifying unpublished drug activities that may account for drugs' therapeutic benefits and/or ADRs. We used the SPD to identify putative target vs. ADR associations via systematic analysis and explain known ADRs via target activities not previously reported in public resources. Beyond the present work, the SPD has broad utility for drug safety and mechanism of action investigations, and phenotypic activity deconvolution for drug activity in cellular models.

## Results

### In vitro safety pharmacology database

In vitro safety pharmacology assays are used to reveal potential clinical ADRs of low molecular weight compounds during lead optimization[8]. To interpret these results in the context of marketed drug safety profiles, we tested 1958 unique drug substances across 200 safety pharmacology assays, resulting in 147 653 concentration-response curves (Supplementary Data 1–3). As some of these results were repeated measurements on different dates and/or different lots of drug substance, results were summarized into 121 097 unique drug vs. assay pairs. The number of assays per drug ranged from 20 to 161, with a median of 66. Thus, SPD represents an unprecedented resource for investigating on- and off-target pharmacology of marketed drugs.

The database was completed over the course of several years. Changes in assay formats (e.g., radioligand binding vs. luminescence), internal vs. external sourcing, and other factors resulted in multiple assays for a given target and mode (agonist, antagonist, inhibitor). To simplify data analysis and maximize the number of tested drugs for a given target and mode, assays were merged into 168 assay groups by analyzing the concordance of repeated measurements. Assay groups ranged from 1 to 4 assays, and often combine similar assays performed internally vs. contract research organizations (CROs). The number of unique drugs tested per assay group ranged from 30 to 1942, with a median of 794 drugs per assay.

In vitro assay results for marketed drugs published in the literature are available in several open-access resources, such as ChEMBL[13] and DrugCentral[14], and from commercial providers of results curated from journals. Results from our database were cross-referenced to these sources, revealing low overall coverage of drug-target pairs, especially for inactive results (Fig. 1a). Further, quantitative agreement of $AC_{50}$ values is modest (Fig. 1b, c), possibly because of heterogeneity in methods used to assess pharmacological activity when aggregated across publications. For drug-assay pairs with results in ChEMBL, 66% of SPD $AC_{50}$ values ≥10 μM have a median ChEMBL $AC_{50}$ <10 μM. Conversely, 82% of SPD $AC_{50}$ values <0.1 μM are reported to be similarly potent in ChEMBL. These observations are consistent with publication bias towards positive or active findings.

To systematically investigate factors contributing to activity differences, we matched SPD vs. 21 596 individual ChEMBL activity results, and annotated each activity pair using assay and target attributes (methods). When modeling differences in log $AC_{50}$ values, SPD attributes denoting agonist assays (Mode), kinase assays (Protein

Class), and protein functional assays (Event, e.g., calcium flux assays) tended to increase differences, as did ChEMBL attributes "protein format" (a Bioassay Ontology term often denoting brain homogenate assays) or ChEMBL standard type of $EC_{50}$ (typically associated with functional assays). SPD attributes denoting binding assays (Mode = Binding or inhibition) were associated with smaller activity differences. As noted above, activity differences tended to be larger when the reported activity was higher in ChEMBL. Notably, comparing assays across species (e.g., human vs. mouse protein) was not associated with larger activity differences. Taken together, these represent received wisdom in comparing assays across sources: assays measuring functional events downstream of targets are more variable than those measuring binding events at targets. These trends are likely to be confounded by the association of measurement approaches and target families difficult to distinguish in our database (e.g., kinase/enzyme assays are cell-free assays and GPCRs or ion channels are cell-based assays).

### Assessing the clinical relevance of drug-assay associations

The clinical relevance of results from in vitro safety pharmacology panels is commonly assessed by calculating a safety margin, or the ratio of in vitro $AC_{50}$ and the therapeutic free plasma concentration at the highest approved dose[5]. To calculate safety margins from the SPD, we compiled human plasma Cmax and plasma protein binding (PPB) results from several sources, obtaining free Cmax estimates for 937 drug substances (Supplementary Data 2; Methods). Across all assays, 6783 drugs vs. assay safety margins were calculated and distinguished by on-target activities (i.e., the assay measures activity at the drug's target), off-target activity that is known (in DrugCentral, ChEMBL, or subscription resources), and non-published off-target activities in the SPD (Fig. 2a). The median on-target margin was 2.4 vs 80 for known off-target activities and 353 for unpublished activities. This suggests that a large proportion of off-target activities from biochemical assays would not manifest as ADRs at clinically relevant exposures.

According to the free-drug hypothesis, biochemical activity from in vitro assays becomes physiologically relevant when the safety margin approaches 1. Overall, 28% (122/429) of on-target margins in our database exceed 10 and 12% (51/429) exceed 50. To investigate the target dependence of safety margins, we tabulated the median value for mechanisms (target and mode; Supplementary Data 4). Among mechanisms represented by ten or more drugs, the median margin ranged from 0.2 (SLC6A4, or serotonin transporter) to 5.9 (HTR2A antagonism). Several mechanisms with lower representation had median margins exceeding 10, and 23/47 mechanisms have 25% or more of drugs with margins of 10 or higher.

We labeled as potentially physiological all off-target activities with a margin of 10 or less, resulting in 517 known and 684 unpublished physiological off-target activities. Drugs with higher overall promiscuity, defined as the percentage of assay groups with $AC_{50}$<10 μM, contributed a significant portion of known and unpublished physiological off-target activities (Fig. 2b). For instance, the promiscuous antidepressant nefazodone (31/88 assays with $AC_{50}$ results <10 μM, or 35%) has four on-target and 23 off-target physiological activities; five off-target activities were not reported in the sources we considered. There are outliers from the overall trend: sunitinib has 51% target promiscuity, yet only a single physiological activity (on-target), owing to its very low free Cmax of 6.3 nM; the antibiotic cefepime has no $AC_{50}$ results <10 μM, yet six off-target activities above this threshold may be physiologically relevant owing to its high free Cmax of 260 μM (Supplementary Fig. 1).

To identify unpublished off-target activities with potential impact on disease unrelated to the on-target activity, we focused on a subset with non-overlapping on- vs. off-target indications according to DrugCentral (Supplementary Data 5). Unpublished activities that were noteworthy included CNR1 inhibition of losartan ($AC_{50}$ = 1.2 nM) and

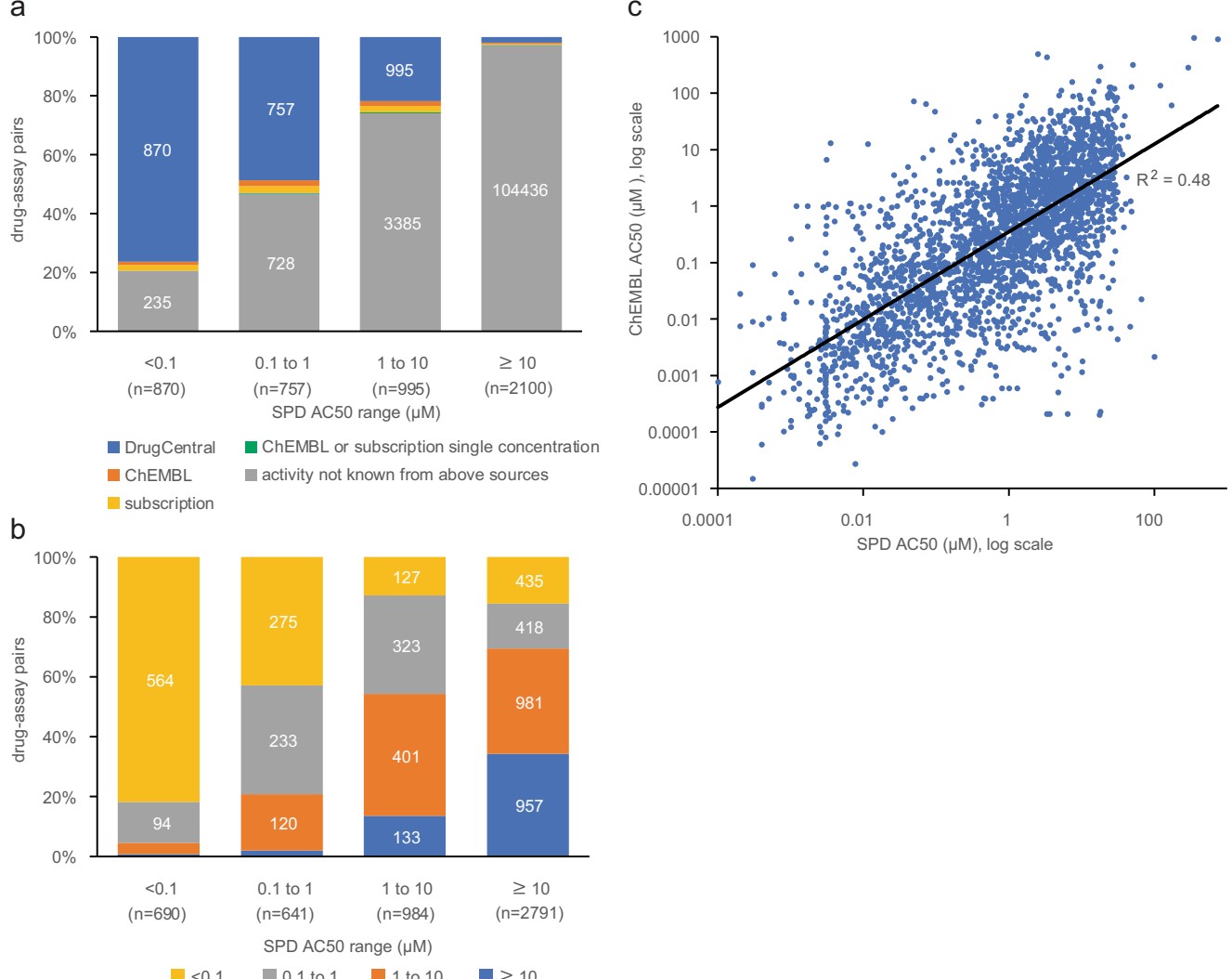

**Fig. 1 | Safety pharmacology results vs. literature-based resources.**
**a** Distribution of $AC_{50}$ values from SPD by $AC_{50}$ range, where highly active results are denoted as $AC_{50} < 0.1 \,\mu M$ and inactive results as $AC_{50} \geq 10 \,\mu M$. Drug-assay pairs were cross-referenced to resources containing results curated from the biomedical literature: DrugCentral $AC_{50} < 10 \,\mu M$ or target annotated as the MOA in DrugCentral (blue), ChEMBL $AC_{50} < 10 \,\mu M$ (orange), subscription resources $AC_{50} < 10 \,\mu M$ (yellow), or single concentration activity $> 50\%$ in either ChEMBL or subscription resources (green). Resources were labeled hierarchically, i.e., activities reported in DrugCentral are mostly available in ChEMBL and other resources. **b** qualitative comparison of median ChEMBL vs. SPD $AC_{50}$ values for 5106 drug-assay pairs; SPD results with $AC_{50}$ qualifier > ($AC_{50}$ greater than max concentration tested) are shown as $\geq 10 \,\mu M$. **c** quantitative comparison of median ChEMBL vs. SPD $AC_{50}$ values for 2700 drug-assay pairs where the SPD $AC_{50}$ qualifier is = (i.e., measurable activity); Pearson $R^2 = 0.48$.

glipizide ($AC_{50} = 19$ nM), which may contribute to their therapeutic effect in treating hypertension and metabolic syndrome (the CNR1 antagonist rimonabant is used in the management of obesity[18]). Rotigotine, a dopamine agonist used for Parkinson's disease, was found to be an ADRA1A ($\alpha_{1a}$) agonist ($AC_{50} = 3.3$ nM); it has been described as an $\alpha_{2b}$ agonist in a journal not curated by the sources used in our analysis[19]. Clinical studies have shown increased systolic blood pressure in treated patients, consistent with $\alpha_{1a}$ agonism[20,21]. Citalopram, a selective serotonin reuptake inhibitor (SSRI), was found to inhibit the histamine H2 receptor (HRH2; $AC_{50} = 350$ nM). Depression[22] and the use of tricyclic antidepressants have been reported to increase the incidence of gastroesophageal reflux disease (GERD), but not SSRI antidepressants[23] (of which citalopram is the most prescribed[24]). Clinical studies suggest direct effects (rather than altered pain perception) on esophageal function[25]. HRH2 antagonists (e.g., ranitidine, cimetidine) reduce gastric acid secretion and are clinically approved to treat GERD symptoms. Therefore, HRH2 inhibition by citalopram may contribute to its observed effects on the digestive system.

Zolpidem, a GABA agonist used for treating insomnia, inhibited CHRM1 ($AC_{50} = 0.21 \,\mu M$), possibly contributing to its observed effect on dystonia[26].

## Evaluation of literature-reported target vs. ADR relationships

Variability in the composition of safety pharmacology assay panels suggests that many target vs. clinical ADR associations are not fully understood[8]. Associations reported in the biomedical literature are summarized in three reviews[5,9,27]. We utilized the SPD to evaluate the significance of associations involving 60 targets, each listed as risk factors for 1 to 34 ADRs coded using MedDRA preferred terms. To characterize the strength of such associations, we correlated drug activity measured in SPD assays vs. clinical ADRs according to SIDER[16] and FAERS, as summarized in DrugCentral[28]. Drug activity was represented as (unadjusted) $AC_{50}$ values, (2) ratio of $AC_{50}$ vs. Cmax, tot (i.e., total margin), and (3) ratio of $AC_{50}$ vs. Cmax, free (free margin), and correlated vs. presence or absence of ADRs using the Kruskal–Wallis (KW) test (Supplementary Fig. 2). Because we performed multiple

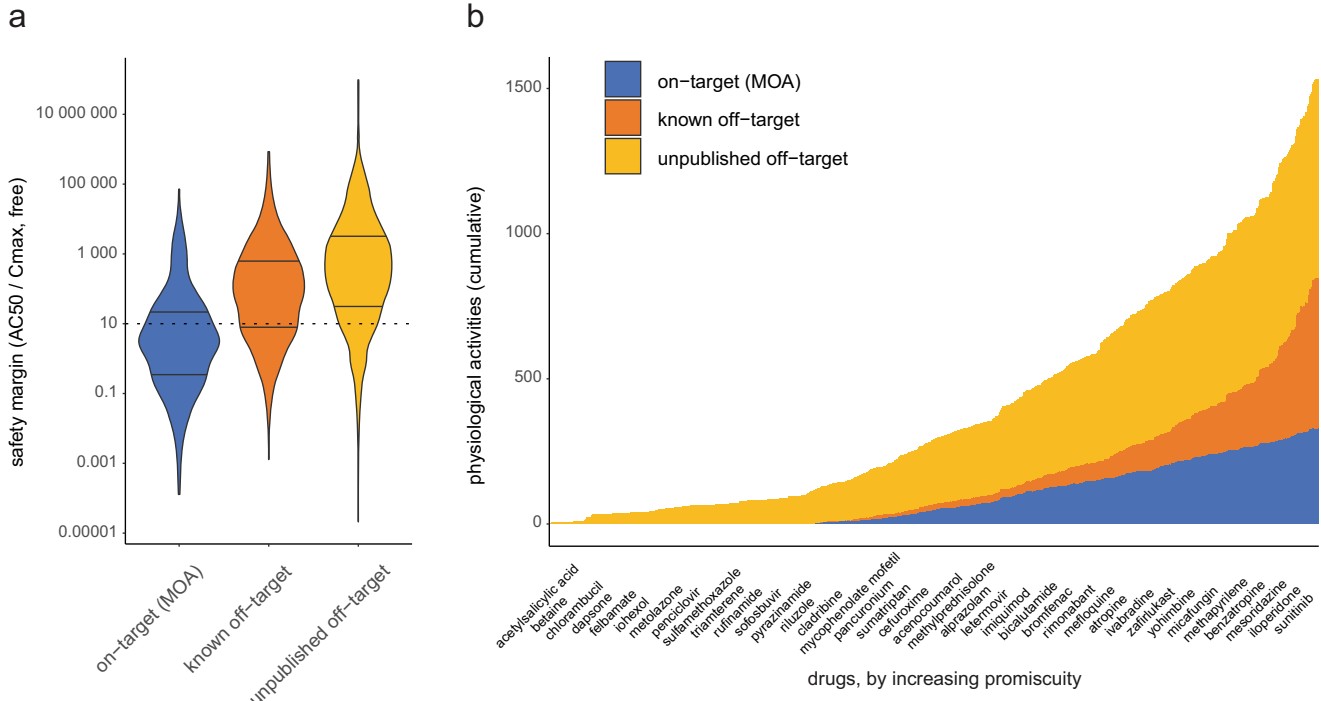

Fig. 2 | **Physiological off-target activities. a** Safety margin ($AC_{50}$ vs. therapeutic free plasma concentration) of drug-assay pairs by type: on-target activities, i.e., the drug's mechanism of action (MOA) (blue), off-target activities known in literature-based resources (orange), unpublished off-target activities (yellow). Distributions are shown as violin plots, with the first and third quartiles denoted with lines. **b** Cumulative distribution of physiological activities (target activities with safety margin <10) for 894 drugs, sorted by increasing overall promiscuity (% of results with $AC_{50} < 10\,\mu M$).

statistical tests per target-ADR pair, associations were classified as significant ($p \leq 0.001$), marginal ($0.001 < p \leq 0.05$), not significant ($p > 0.05$) or not tested (associations with fewer than 10 positive or 50 negative drugs for the ADR). We also imposed a minimal threshold ROC AUC $\geq 0.6$ for distinguishing positive vs. negative drugs for a given ADR (Fig. 3a). Across 719 tested associations, 240 (33%) were significant and a further 20 (3%) were marginal. The proportion varies significantly across targets (Fig. 3b). A large percentage of associations were confirmed for adrenergic receptors (e.g., ADRA1A; 15/15), muscarinic receptors (e.g., CHRM1; 30/34), 5-HT receptors (e.g., HTR1A; 20/20−the notation indicates number significant + marginal/number tested). Our evaluation of target-ADR relationships from Bowes et al.[5], which represents a consensus of safety pharmacology targets across several pharmaceutical companies, is summarized in Supplementary Table 1, with full results in Supplementary Data 6. Similar results were obtained using alternate FAERS risk or assay score thresholds (Supplementary Notes).

Several targets had no statistically significant associations, despite many potential ADRs reported in the literature. Lack of significance might be due to characteristics of the assay data (i.e., few actives, low statistical power), few drugs causing an ADR, biases towards certain ADR types, etc. To investigate further, associations were labeled as significant ($p \leq 0.001$ and ROC AUC $\geq 0.6$) or non-significant (all others). We created a Lasso-penalized logistic regression model of these outcomes using several properties, including measures denoting the proportion of drugs active in the assay and MedDRA system organ class (SOC) of the ADRs. A small number of variables, including decreasing 5th percentile of $AC_{50}$ values, increasing count of drugs with $AC_{50} < 1\,\mu M$, and ADRs belonging to the SOCs "Nervous system disorders" or "Psychiatric disorders" all increased the probability of a target-ADR association being significant (Supplementary Fig. 3). These results are intuitive: targets having many drugs with potent $AC_{50}$ values (the percentile measure), or sub-micromolar actives, and CNS-related ADRs observed when modulating promiscuous GPCRs[5] are more likely to have significant ADRs.

The model class assignments ("likely significant" or "likely non-significant") can be viewed as a prior likelihood of validating a target-ADR association, given the characteristics of the assay data and the ADR class. When evaluating the predictions for 459 literature associations having $p > 0.05$ or ROC AUC <0.6, 414 (90%) were assigned the likely non-significant class. These are literature-reported associations identified by the model as having a low likelihood of being significant, given the dataset. Conversely, 45 associations with $p > 0.05$ were identified by the model as likely significant (i.e., dataset characteristics should enable validation of a true association). These include both serious ADRs (heart failure for ADRA2B, DRD1, and DRD2 activation) and lower severity effects (sleep or memory impairments for several targets).

Safety margins, either based on free[5] or total Cmax[29], have been proposed as superior to unadjusted $AC_{50}$ for predicting ADRs. A practical limitation is that estimated human Cmax is usually not available in early lead optimization. The proportion of significant associations was highest when using $AC_{50}$ as an activity measure, more notably when using SIDER as the source of ADR annotations (Supplementary Fig. 4). This remained true when using the ROC AUC rather than $p$ value for selecting significant associations, a measure which should be less sensitive to the smaller sample sizes for margin-based activity measures. Even for target-ADR pairs significant on both $AC_{50}$ and free margin, the strength of association is generally higher for $AC_{50}$-based activity measures (Supplementary Fig. 5).

**Systematic evaluation of target vs. ADR relationships**

The 743 literature-derived targets vs. ADR associations represent a small proportion of all possible such associations. We systematically evaluated all possible target vs clinical ADR annotations from SIDER and FAERS. To increase the power to detect associations for less common ADRs, we also modeled assay vs. MedDRA high terms (HT) and group terms (HG) that combine related preferred term (PT) ADRs. In total, we examined 562 744 relationships for each of 124 assays and

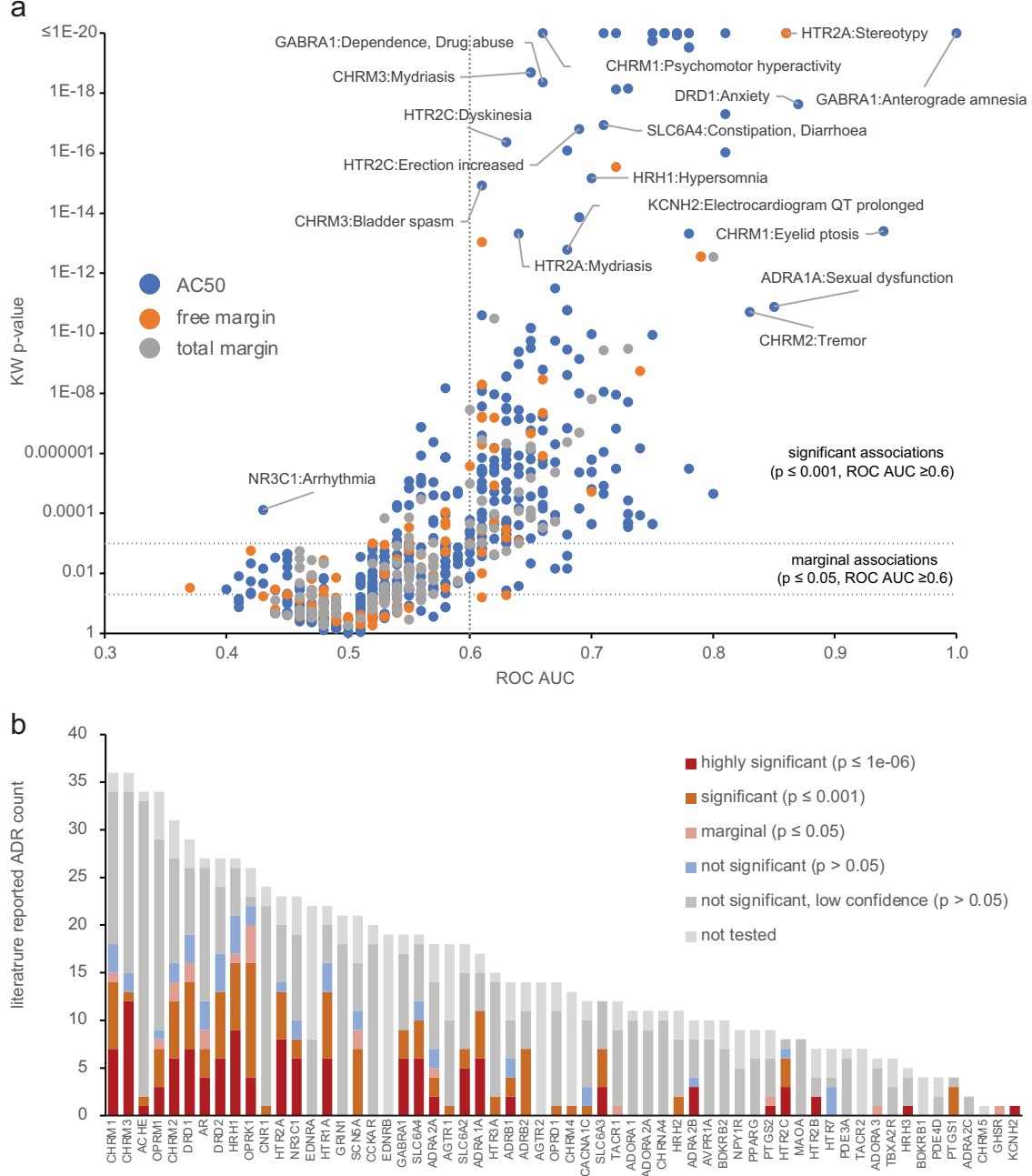

**Fig. 3 | Statistical significance of literature-reported target vs. ADR associations. a** Assessing significance by comparing Kruskal–Wallis (KW) *p* value vs. receiver operator characteristic area under the curve (ROC AUC) for 884 associations, using the activity measure $AC_{50}$ (blue), free margin (orange) or total margin (gray) with a smallest *p* value for each association; regions considered significant ($p \le 0.001$ and ROC AUC $\ge 0.6$) or marginal ($p \le 0.05$ and ROC AUC $\ge 0.6$) are indicated within dashed lines; a small number of associations having $p \le 0.05$ and ROC AUC $< 0.5$ denote associations where the direction is opposite to that reported in the literature (drugs with activity have a lower probability of exhibiting the ADR). **b** Total number of ADRs reported across targets, distinguished by the level of statistical significance observed.

2647 MedDRA terms with sufficient representation in SPD, using three activity measures for each pair ($AC_{50}$ values in μM, total margin, free margin). Overall, 1992 assay vs. MedDRA pairs met the statistical criteria of ROC AUC $\ge 0.7$ and KW *p* value $<1e-06$ on one or more activity measures (Supplementary Data 7). This included 671 associations using HT or HG terms, of which 560 had one or more significant child terms. To limit redundancy, we focused further analysis on 1321 PT and 111 HT/HG-based associations without a more predictive child term.

Only 25% of the 1432 assay vs. ADR associations overlapped with those reported in the literature reviews, suggesting that some might be worthy of investigation (Fig. 4a). Mirroring our findings for literature associations, a large proportion were significant only on $AC_{50}$, compared to margin-based activity measures (Fig. 4b). However, we noted that the proportion supported by a margin-based measure increased with the proportion of active drugs that were on-target, suggesting that margin-based associations were more likely to be plausible (Fig. 4c).

Drugs often modulate off-targets with high sequence similarity (and hence similar binding pockets) vs. the intended target. Statistically significant off-target vs. ADR relationships from the univariate analysis may be confounded by the corresponding on-target relationship. For example, the most significant association involving

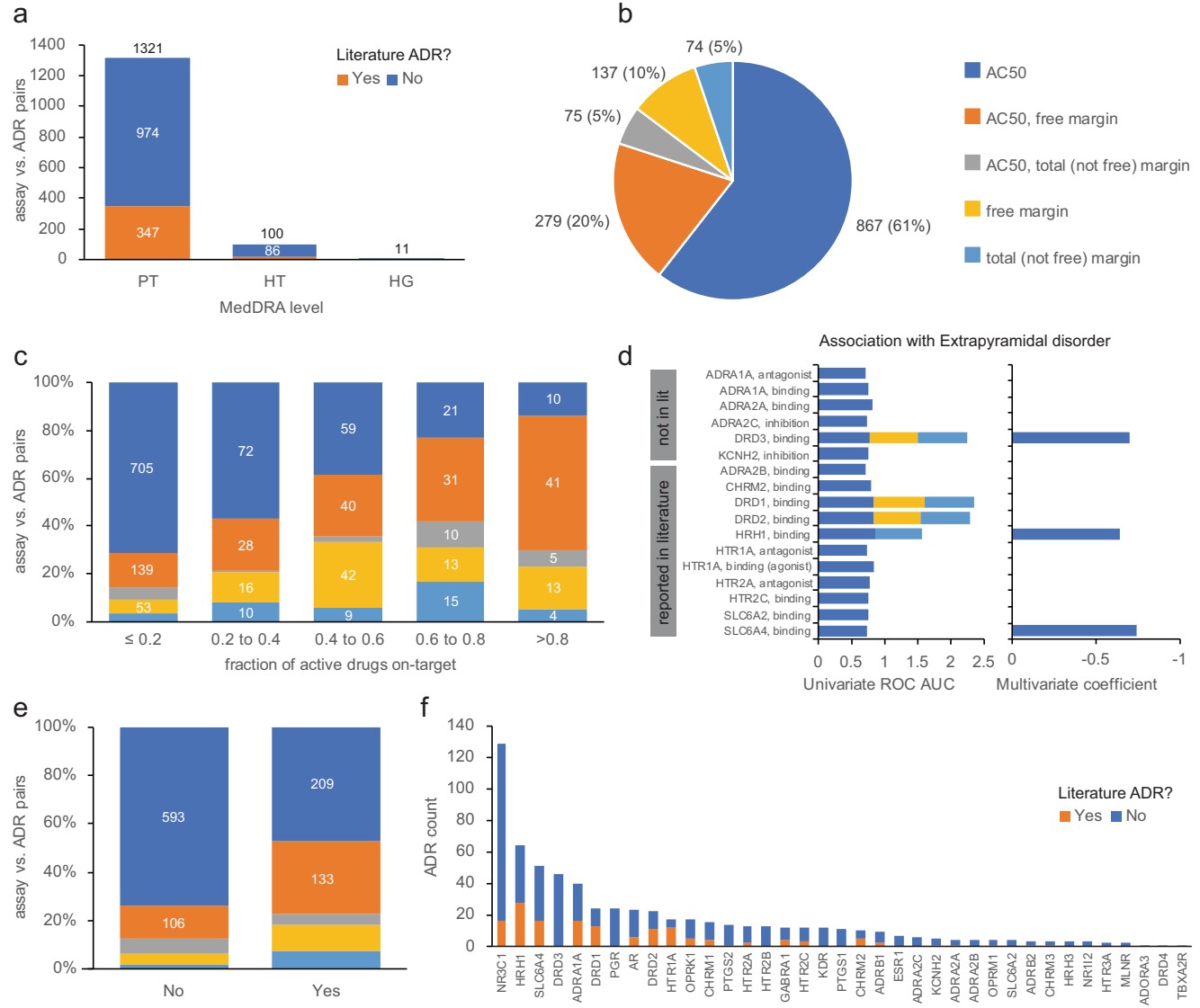

**Fig. 4 | Systematic identification of assay vs. ADR relationships. a** Number of assay vs. ADR pairs having KW *p* value ≤1e-06 and ROC AUC ≥0.7 in either SIDER or FAERS, by MedDRA type (PT preferred term, HT high term, HG group term), distinguished by their presence or absence among the literature-reported associations. **b** Proportion of associations considered significant by three activity measures considered; associations significant by free margin alone, or both free and total margin, are labeled as "free margin"; associations significant by a total margin only are labeled as "total (not free) margin". **c** Distribution of associations by activity measure vs. fraction of drugs active in the assay that are on-target (i.e., activity at the drugs' known targets). **d** Identification of non-redundant assays linked to "extrapyramidal disorder". The left panel indicates univariate ROC AUC for each assay, showing cumulatively in stacked form significant associations for AC$_{50}$ (blue), free margin (yellow), and total margin (cyan). The right panel indicates the coefficient in the penalized logistic regression model, i.e., only 3 assay AC$_{50}$ coefficients have non-zero values and are therefore considered non-redundant in explaining ADR risk. **e** Comparing assays with non-zero vs. zero coefficients in the logistic model by the proportion of significant activity measures. **f** Distribution of 631 non-redundant assays vs. ADR pairs by target, distinguished by their presence or absence among the literature associations.

glucocorticoid receptor (NR3C1) binding vs. "Adrenal cortical hyperfunctions" (MedDRA 10001341; *p* value = 2e-42, ROC AUC = 0.87) is mirrored by a similar relationship for progesterone receptor (PGR) agonism (*p* value = 1e-17; ROC AUC = 0.87). Among 12 drugs causing this ADR, according to SIDER and tested in both assays, eight drugs modulate both NR3C1 and PGR with AC$_{50}$ <10 μM; seven drugs are glucocorticoid antagonists, and one drug is a PGR agonist. Achieving selectivity is difficult[30], and whether one or both targets contribute to the occurrence of this ADR cannot be determined via univariate statistical analysis.

To distinguish overlapping vs. orthogonal assay contributions to clinical ADR risk, we performed multivariate logistic regression to model the probability of observing a given ADR using the subset of significant assays identified above. Because unpublished high-significance assays and known lower-significance assays might explain the same variation in ADR risk, we included assays for literature targets that reached the lower-significance threshold *p* ≤ 0.001 (i.e., significant per Supplementary Data 6 but not reaching the combined *p* ≤ 1e-06 and ROC AUC ≥0.7 for inclusion in Supplementary Data 7). The analysis workflow employing the lasso penalty seeks to select the smallest number of variables (assays), resulting in a model with predictive accuracy within one standard error of the best model (with any number of variables). Accordingly, assays retained in the sparse model are likely to represent distinct contributions. The selection of assays retained in the sparse models was stable across random resampling of the data (Supplementary Notes).

To illustrate, two datasets were modeled to identify non-redundant assays for predicting "Adrenal cortical hyperfunctions": one from SIDER consisting of 487 drugs annotated as positive (18) vs negative (469) for the ADR, and a second from FAERS consisting of 605 drugs (17 positives vs 588 negatives). These datasets are not identical owing to differences in SIDER and FAERS, with 11 shared positive drugs in both datasets. Measures of $AC_{50}$, total margin and free margin for NR3C1 and PGR were employed as predictors (3 × 2 variables). The optimal (sparse) model for each dataset was reduced to a single variable, namely the NR3C1 $AC_{50}$. This suggests that the total and free NR3C1 margins and the PGR endpoints explain the same variation in the ADR risk as the NR3C1 $AC_{50}$. Put differently, there is no evidence of utility beyond the NR3C1 $AC_{50}$ to predict this ADR. For "extrapyramidal disorder" (MedDRA 10015832) SIDER annotations, this approach reduced the number of predictive assay + activity measure pairs from 28 to 3, namely SLC6A4, HRH1, and DRD3 binding $AC_{50}$ values (Fig. 4d); DRD3 is not reported as a risk factor for this ADR according to the literature reviews. Overall, this approach eliminated 801 assay vs. ADR pairs identified as significant by univariate analysis but redundant with other retained (more predictive) assays. Comparing retained vs. eliminated assays shows enrichment towards margin-based measures (Fig. 4e).

In summary, we identified 189 ADRs with a single predictive assay ($p$ value ≤1e-06 and ROC AUC ≥0.7; no multivariate modeling) and a further 442 ADR vs. assay pairs with univariate significance and retention in the sparse models. The glucocorticoid receptor (NR3C1),
histamine H1 receptor (HRH1), serotonin transporter (SLC6A4), dopamine D3 (DRD3), adrenergic alpha-1A (ADRA1A), and dopamine D1 (DRD1) receptors accounted for the largest proportion of ADRs (Fig. 4f).

## Investigation of unpublished target vs. ADR relationships

The analytical workflow described above systematically evaluated all possible assay (target) vs. ADR relationships, identifying 631 associations that met stringent statistical criteria ($p$ value ≤1e-06 and ROC AUC ≥0.7) and were not redundant with known risk factors (i.e., retained via the sparse modeling). As validation of this approach, 149 associations (24%) were from the literature reviews described above. This suggests that a subset of the remaining 482 associations may represent clinically relevant unpublished target vs. ADR risk factors.

Amongst these, the glucocorticoid receptor (NR3C1) accounted for many associations. Even though these were not all listed in the literature reviews, immune suppression that results from activity at NR3C1 is well recognized[31], and most of these interactions are on-target. Activity linked to the serotonin transporter (SLC6A4) similarly shows a range of ADRs associated with the indication of mood disorders that are treated with SSRI drugs[32].

We searched the literature for associations where 20% or more of active drugs were off-target and that seemed plausible to us (Table 1). For example, ADRA2C inhibition was associated with auditory hallucination and paranoia in FAERS, on $AC_{50}$ and margin

**Table 1 | Selected assays vs adverse drug reaction relationships from univariate and multivariate analyses with support in the biomedical literature**

| Assay name | MedDRA name | KW $p$ value | ROC AUC | significant sources [a] | Literature [b] |
|---|---|---|---|---|---|
| ADRA1A binding | Eosinophilia | 3.6E-08 | 0.85 | F | 32194050 |
| | Tardive dyskinesia | 1.9E-23 | 0.87 | S | 45, 46 |
| ADRA2C inhibition | Hallucination, auditory | 1.1E-11 | 0.86 | F | 33–35 |
| | Paranoia | 1.2E-09 | 0.82 | F | 33–35 |
| AR binding (agonist) | Erythema nodosum | 2.6E-10 | 0.70 | S | 27075133 |
| | Ovarian and fallopian tube cysts and neoplasms | 2.4E-09 | 0.70 | S | 30791431 |
| | Retinal embolism and thrombosis | 6.5E-07 | 0.74 | S | 29040227 |
| | Ovarian neoplasms malignant 55(excluding germ cell) | 5.9E-12 | 0.82 | S | 30791431 |
| CHRM1 inhibition | Micturition urgency | 4.0E-09 | 0.73 | S | 35117285 |
| | Dental caries | 3.9E-07 | 0.73 | S | 31289718, 12974516 |
| CHRM2 binding | Ileus paralytic | 2.7E-10 | 0.76 | S | 14607264 |
| DRD2 binding | Breast enlargement | 2.4E-08 | 0.72 | S | 36–38 |
| | Menstruation irregular | 2.4E-08 | 0.71 | S | 38 |
| | Salivary hypersecretion | 7.9E-11 | 0.71 | S | 16421461 |
| DRD3 binding | Tardive dyskinesia | 4.3E-31 | 0.90 | F, S | 40–42 |
| | Extrapyramidal disorder | 5.1E-33 | 0.87 | F, S | 19506579 |
| | Gambling disorder | 2.8E-09 | 0.83 | S | 26192187 |
| | Hyperprolactinemia | 1.6E-17 | 0.86 | S | 16169407, 33854317 |
| | Blood prolactin increased | 4.2E-14 | 0.88 | S | 33854317 |
| HRH1 binding | Hypomania | 5.0E-17 | 0.88 | S | 34572558 |
| HTR2B antagonist | Lethargy | 7.4E-07 | 0.71 | S | 30666218 |
| | Sleep disorders NEC | 1.9E-08 | 0.72 | S | 30666218 |
| OPRK1 binding | Salivary hypersecretion | 2.8E-08 | 0.74 | S | GO[c] |
| PGR agonist | Hyperpigmentation disorders | 7.8E-13 | 0.71 | S | 39 |
| SLC6A4 binding | Generalized tonic-clonic seizure | 1.5E-08 | 0.76 | F | 31849820 |
| | Galactorrhea | 2.0E-21 | 0.76 | S | 14997175 |
| | Blood prolactin increased | 2.5E-07 | 0.78 | S | 14997175 |

[a] Sources are abbreviated as S (SIDER) and F (FAERS).
[b]References not discussed in the text are provided as PubMed IDs.
[c]Evidence from Gene Ontology.

measures. A mouse knockout model showed increased startle reflex and aggression[33]. Differential gene expression[34] and genetic alterations[35] in ADRA2C also suggest a role in schizophrenia. DRD2 and DRD3 binding were associated with mammary and menstruation-related ADRs in SIDER[36–38]. PGR agonism was associated with hyper-pigmentation disorders in SIDER, with drugs acting both on-target (medroxyprogesterone and progesterone) and off-target (NR3C1 modulators). A recent report describes the effect of asoprisnil, a selective PGR modulator, on melanocytes[39].

More challenging is the interpretation of motor dysfunction ADRs associated with modulation of DRD3 and adrenergic receptors, given the preponderance of evidence implicating DRD1 and DRD2 (Fig. 4d). Even though sparse model building selected DRD3 over DRD2, both targets may explain the same variation in ADR risk. DRD3 gene polymorphisms have been associated with tardive dyskinesia (TD)[40–42], and DRD3 knockout animals have slightly altered locomotor activity, and enhanced sensitivity to DRD1/DRD2 agonists[43]. Motor dysfunction-related ADRs are listed in the FDA label for cariprazine, an atypical antipsychotic selective for the D3 receptor[44]. As such, DRD3 may contribute to the ADR profile of dopaminergic drugs, usually ascribed to DRD2 in the literature. Although TD is strongly linked with abnormalities of the dopaminergic system, there is some evidence from clinical case studies, that 3H-dihydroergocryptine (3H-DHE)-alpha2 adrenergic receptor binding and cerebrospinal fluid nor-epinephrine (NE) were positively correlated with the severity of TD[45,46]. It remains elusive to link drug-induced TD to engagement with adrenergic receptors by antipsychotic drugs, as most of them have high pharmacological promiscuity with prominent effects at dopamine, histamine, and serotonin receptors, also associated with movement disorder side effects. However, it is plausible that combined effects at these receptors include the adrenergic component.

## SPD assay results reveal the putative cause of known drug ADRs

Many drug vs. assay results from the SPD are not described in the pharmacology resources we interrogated. To determine if these unpublished activities explain known ADRs of marketed drugs, we tabulated 325 drug-target-ADR triples for target-ADR relationships reported in the literature and statistically significant in our analysis (Fig. 5 and Supplementary Data 8, 9). The most prevalent class of newly explained ADRs were cardiac and respiratory effects. These include 16 drugs active at OPRK1 with association "cardio-respiratory arrest", with $AC_{50}$ values ranging from 0.35 to 4 µM. A variety of ADRs belonging to movement disorders (e.g., extrapyramidal disorder, Parkinsonism, and dyskinesia) were associated with several targets, including OPRK1 (9 drugs), CHRM2 (8 drugs), and ADRA1A (5 drugs). We investigated selected examples in further detail (Table 2).

## Case study: accommodation disorder and muscarinic activity

Accommodation disorder is one of the most common ADRs associated with muscarinic receptor antagonists[47]. We found several drugs that do not have known association with muscarinic receptors, but caused accommodation disorder. Zolpidem is a GABA-A receptor agonist with no muscarinic receptor-related side effect in its label[48]. However, FAERS data suggest that it is related to accommodation disorder, clearly associated with muscarinic receptor antagonism but not with GABA. When we tested zolpidem for its effects at a large range of targets, we found that in addition to engagement to the GABA-A receptor ($AC_{50}$ = 46 nM) it also bound to the M1 muscarinic receptor with high potency ($AC_{50}$ = 210 nM). Eletriptan is a highly selective HTR1A receptor agonist for the treatment of migraine[49]. To our knowledge, there is no evidence that it has pharmacodynamically relevant muscarinergic engagement, which could explain the predicted accommodation disorder in association with the M2 muscarinic receptor.

## Case study: citalopram off-target-ADR associations

Citalopram is an SSRI antidepressant used to treat anxiety disorders and other psychiatric conditions[50]. While catecholamine uptake was broadly investigated with SSRIs[51], there is little knowledge about their engagement with monoaminergic and other CNS receptors, in contrast to other antidepressants[52]. We found several ADRs that could be associated with dopaminergic engagement: movement or extra-pyramidal disorders, psychotropic disorders, and endocrine adverse reactions (Supplementary Data 8). Our results indicate modest activity at D1 and D3 ($AC_{50}$ = 4 and 5 µM, respectively), but not D2 receptors ($AC_{50}$ > 10 µM). The major metabolite of citalopram, desmethyl-cita-lopram, has been reported to have similar binding activity at the D3 receptor[53].

There is published evidence that SSRIs, including citalopram, are rarely associated with TD. The present hypothesis is that this is related to an indirect anti-dopaminergic effect induced by increased levels of serotonin[54,55]. However, there is indirect support for the interaction of citalopram with the dopaminergic system. Citalopram induces the upregulation of dopamine D1, D2, and D3 receptor mRNA levels in the rat nucleus accumbens[56], which is an essential part of the rewarding and primary motivational CNS network. The results indicate that alterations in the availability of neurotransmitters at synapses induced by citalopram are strong enough to induce immediate and long-lasting adaptive changes in the neuronal network. However, the strongest effect was observed with the D2 receptor[57]. This finding also raises the possibility, that homo- and/or hetero-dimerization[58] of the dopamine receptors might occur due to citalopram treatment and possibly result in D2 upregulation[59], suggesting that antidepressants can induce adaptive changes in the brain.

Based on the engagement with the dopaminergic system, it is not surprising that citalopram also has psychotropic ADRs similar to that of atypical antipsychotic drugs. However, this class of ADRs is difficult to differentiate from symptoms associated with the treated disease[12]. Also, the serum concentration of citalopram is affected by con-comitant treatment with neuroleptics, benzodiazepines, and tricyclic antidepressants and by the age of the patients[60]. These conditions have two important effects on the ADR association of citalopram, namely the concomitant treatment could cause the side effects and the ele-vated concentration of citalopram could bring its weak effects at the serotonin and dopamine receptors in coverage. Thus, careful analysis is needed to link common ADRs of psychotropic drugs to citalopram.

Finally, we have encountered information on hormonal changes, namely hyperprolactinemia, an endocrine disorder that is associated with risperidone, an atypical antipsychotic drug[61]. It manifests in galactorrhea and gynecomastia, particularly prominent in boys treated for irritability associated with autism[62]. In a previous study, we linked a high volume of hyperprolactinemia/gynecomastia reporting in FAERS to the strong engagement of risperidone to serotonin and dopamine receptors, transporters with a narrow safety window[12]. However, endocrinal ADRs are rarely associated with antidepressant drugs (SSRIs) and monoamine oxidase inhibitors (MAO-I)[63]. The HRH2 activity noted herein may therefore contribute to the ADR of hyper-prolactinemia reported in SIDER.

## Discussion

In vitro safety pharmacology assays are an important tool in lead optimization and risk assessment prior to human studies. With the goal of interpreting results for new chemicals in context, we created the Secondary Pharmacology Database (SPD) over a multi-year period to characterize approximately 2000 drugs. In this work, we present a comprehensive analysis of bioactivity vs. ADR relationships using uniform and standardized assay protocols. Many of these assays are offered by commercial vendors, allowing the application of the SPD for probing the safety characteristics of newly synthesized compounds.

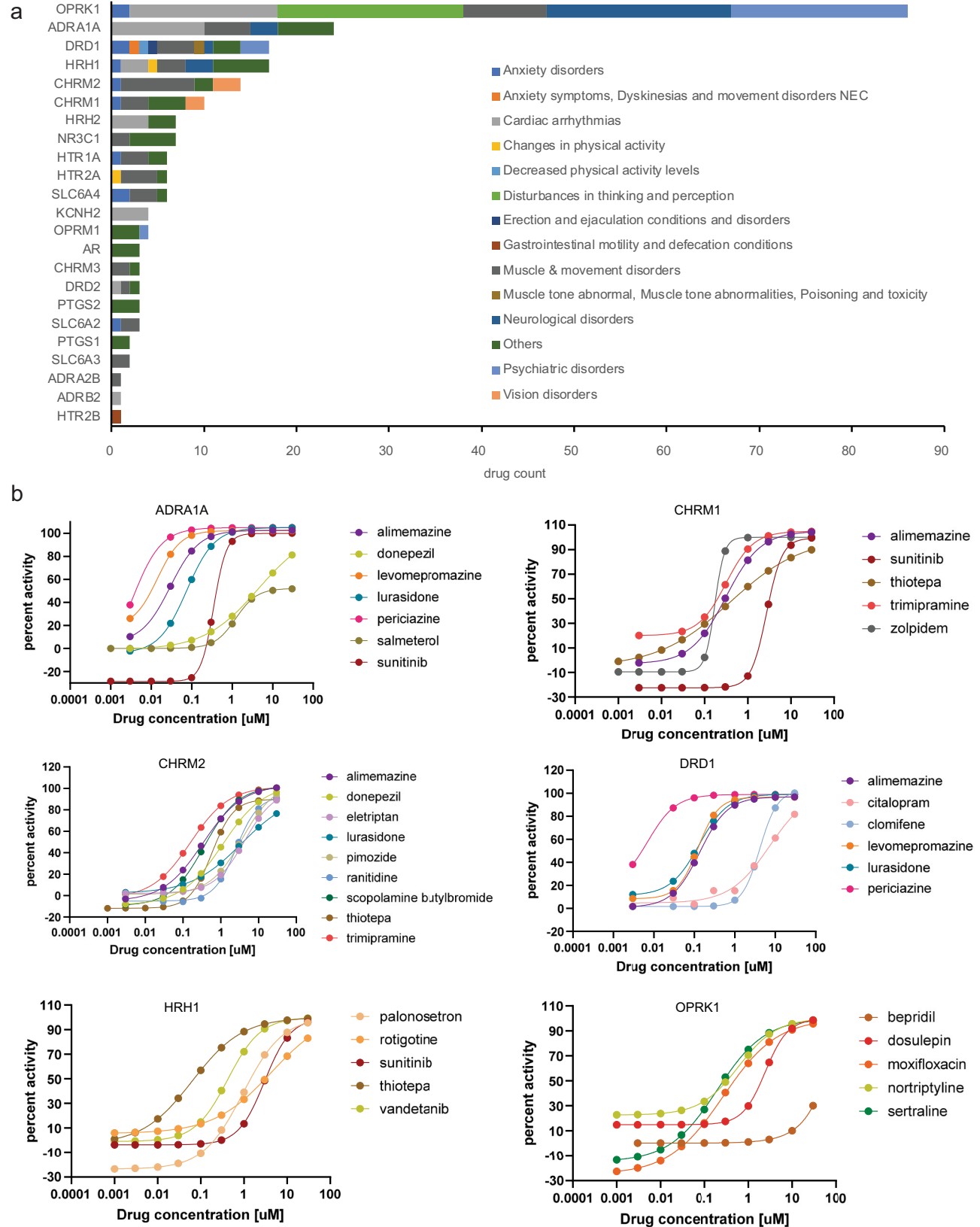

**Fig. 5 | Unpublished off-target activity as putative mechanisms for known drug vs. ADR associations. a** Number of drug vs. ADR associations with unpublished off-target activity by the target. **b** Concentration vs. activity for selected drugs and assays.

**Table 2 | Selected adverse drug reaction newly explained by SPD database**

| Drug (targets) | Assay target | AC$_{50}$ (μM) | Explained ADRs |
|---|---|---|---|
| amiodarone (KCNH2) | HRH1 | 2.4 | akinesia, bradykinesia, decreased activity, disorientation, extrapyramidal disorder, hypokinesia, inappropriate antidiuretic hormone secretion |
| amlodipine (CACNA1C) | ADRB2 | 1.8 | sinus bradycardia |
| | DRD2 | 2.9 | parkinsonism |
| | OPRK1 | 1.2 | cardio-respiratory arrest, drug abuse |
| citalopram (SLC6A4) | DRD1 | 3.9 | akathisia, catatonia, choreoathetosis, dyskinesia, ejaculation disorder, extrapyramidal disorder, libido increased, movement disorder, orgasmic disorders and disturbances, parkinsonism, priapism, psychotic disorder, serotonin syndrome, substance related and addictive disorders, tardive dyskinesia, torticollis, trismus |
| citalopram (SLC6A4) | HRH2 | 0.35 | hyperprolactinemia |
| donepezil (ACHE) | ADRA1A | 4.4 | drooling |
| | CHRM2 | 1.1 | extrapyramidal disorder |
| eletriptan (HTR1B, D, F) | CHRM2 | 4.2 | accommodation disorder |
| fluoxetine (SLC6A4) | OPRK1 | 1.5 | cardio-respiratory arrest, coordination abnormal, delusion, drug abuse |
| lurasidone (DRD2, HTR2A) | ADRA1A | 0.075 | blood prolactin increased, cogwheel rigidity, drooling, muscle rigidity, oculogyric crisis, sexual dysfunction, torticollis |
| | CHRM2 | 3.5 | dyskinesia |
| | DRD1 | 0.13 | dyskinesia, sedation, substance related and addictive disorders |
| pregabalin (CACNA2D1,2) | ADRA1A | 3.7 | autonomic nervous system imbalance, blood prolactin increased, dizziness postural, miosis, orgasmic disorders and disturbances, restlessness, sudden death, torticollis, trismus, urinary incontinence |
| quetiapine (DRD2, HTR2A) | OPRK1 | 1.6 | cardio-respiratory arrest, choreiform movements, coordination abnormal, delusion, drug withdrawal syndrome |
| | OPRM1 | 1.8 | drug dependence, respiratory arrest |
| ranitidine (HRH2) | CHRM2 | 2.6 | dyskinesia |
| sibutramine (SLC6A2,3,4) | HTR2A | 1 | torticollis |
| | OPRK1 | 4.8 | delusion |
| tramadol (OPRM1) | CHRM2 | 1.5 | dyskinesia, parkinsonism |
| trimipramine (SLC6A2,3,4) | CHRM1 | 0.31 | accommodation disorder, mydriasis |
| | CHRM2 | 0.16 | accommodation disorder, extrapyramidal disorder, mydriasis, parotid gland enlargement |
| valbenazine (SLC18A2) | HTR1A | 1.4 | tardive dyskinesia |
| zolpidem (GABRA1) | CHRM1 | 0.21 | accommodation disorder, mydriasis |

By comparing results from our database to freely available (ChEMBL, DrugCentral) and subscription resources, we found that ca. 95% of assay results were unique to the SPD. This proportion was highest among inactive results but remained ca. 36% for results with AC$_{50}$ < 1 μM. When activity results in SPD were also described in the public resources from literature curation, literature-reported AC$_{50}$ values tended to be smaller (i.e., more potent). These observations are consistent with a bias toward publishing active results.

The availability of inactive results (typically AC$_{50}$ ≥ 30 μM) for many drug vs. target pairs allowed us to test the association between the presence or absence of drug ADRs and the presence or absence of activity at a given target. When evaluating the statistical significance of literature-reported target vs. ADR relationships, we found 64% with no support. This varied significantly by target, suggesting that smaller panels of assays based on carefully selected targets could be used at earlier stages of lead optimization. Our modeling suggested that many associations lacking support reflect limitations in the dataset (i.e., too few potent drugs for the target). However, 128 associations flagged by the model as unlikely to be significant have strong support from our data ($p < 0.001$). This suggests that many of the literature associations lacking support may have modest predictive utility (low effect size), even if substantiated in larger datasets. Contrary to our expectations, we found more relationships to be supported by unadjusted AC$_{50}$ values vs. human Cmax-derived safety margins. From a practical perspective, this facilitates risk assessment early in lead optimization when human Cmax estimates may not be available.

We systemically analyzed target vs. ADR pairs using our database, identifying unpublished associations. Statistical analyses linking targets to ADRs are highly confounded by polypharmacology, whereby ADR risk might be attributed to target activity correlated with the causal risk drivers. Prior investigations using publicly available assay data either did not control[15] or clustered similar associations to identify putative drivers[10]. We attempted to control known associations by performing penalized multivariate selection using all individually significant assays, including lower-significance associations reported in the literature. This approach eliminated ca. 800 associations, increasing the likelihood that those retained are causal.

Methods used in this work to annotate drugs with ADRs have limitations. Expert annotation of structured product labels is limited to small drug sets[64], and revealed limitations of natural language processing (SIDER) or post-marketing spontaneous reports as approximations. Severity and frequency are not generally available in SIDER, and hence not reflected in the associations described herein. Inferring ADRs from FAERS has several pitfalls, including reporting biases and confounding by drug indication[12]. Various statistical approaches have been proposed to extract trends from FAERS data[65–67], and consensus on the most effective approach remains elusive. Our work leveraged the likelihood ratio test[68] as implemented in DrugCentral; other approaches may yield different results.

Since many druggable proteins were not included on our panels, the absence of the causal proteins would fail to deselect non-causal proteins. For instance, 12 associations involving the protein kinase KDR are listed as unpublished in Supplementary Data 7. These include "stomatitis", "gastrointestinal perforation", and "malignant neoplasm progression". These associations may represent general

kinase-mediated ADRs, with the latter illustrating confounding of ADRs due to drug treatment vs. symptoms associated with the treated disease.

An illustrative example concerns associations between targets and "electrocardiogram QT prolonged". Logistic regression was applied to several assays, which are either described as risk factors for long QT in the literature reviews (ADRB2, HRH1, KCNH2, and SLC6A2) or were identified by univariate analyses (ADRA1A, DRD2, DRD4, HRH2, and HTR2B). Among the literature-reported associations, activity at the hERG channel (KCNH2) is thought to be the primary risk factor. The penalized modeling approach retained ADRA1A, DRD2, HRH1, and hERG as non-redundant variables. The role of all but hERG is controversial. Several drugs devoid of activity at hERG (i.e., $AC_{50} > 10\,\mu M$) are annotated with long QT in SIDER or FAERS and support these associations (i.e., $AC_{50} < 1uM$): ADRA1A (alfuzosin, clonidine, mianserin, olanzapine, quetiapine), DRD2 (amisulpride, olanzapine, quetiapine), and/or HRH1 binding (cetirizine, mianserin, mirtazapine, olanzapine, quetiapine, and valproic acid).

There is overwhelming evidence of cardiac arrhythmias caused by H1 histamine receptor antagonists. While a broad range of antihistamines could cause QT prolongation either on their own or in combination with other drugs, there has been an emerging common denominator, which makes this class of medications prevalent to cause cardiac arrhythmias. That is hERG channel inhibition, which has been reviewed extensively during the past two decades[69]. All H1 antihistamines and/or their metabolites—with very few exceptions—have a direct hERG effect with various levels of potency. Their effect might be exacerbated by high exposure because of the co-administration of drugs interfering with the metabolism of the antihistamines. Alternatively, combination therapy with other hERG-inhibiting drugs could synergize their effects. In summary, caution is needed when ubiquitous off-target effects appear in a class of drugs aiming at the same therapeutic target.

Throughout this work, we labeled as unpublished off-target any activities in SPD not reported in ChEMBL, DrugCentral or the subscription resources containing curated pharmacological activity results. Curation-based resources have limited journal coverage, and results we claim as unpublished can sometimes be found with manual searches (e.g., PubMed, Google Scholar). For example, lurasidone and vandetanib are annotated with the ADR "Torsade de pointes" and were found to have hERG $AC_{50}$ values of 0.53 and 0.35 μM, respectively. These activities are not reported in the sources we considered, but are reported in the literature[70,71]. A condition for practical large-scale analyses of pharmacological activity results is inclusion in commonly used databases. As such, activity results from SPD are a significant addition to existing resources summarizing the bioactivity of drugs.

## Methods

### Compliance with ethical regulations
Human clinical data on adverse drug reactions were obtained from publicly available resources that contain results in aggregated form only. These resources do not provide individually-identifiable healthcare information under the Health Insurance Portability and Accountability Act of 1996 (HIPAA). As such, no institutional review board approval was required to use these resources.

### In vitro safety pharmacology assays
Compounds were obtained from the Novartis Institutes of Biomedical Research (NIBR) compound library and tested in a panel of in vitro biochemical and cell-based assays at Eurofins and/or NIBR in concentration-response (eight concentrations, half-log dilutions starting at 10 or 30 μM). Assay formats varied from radioligand binding, to isolated protein, and cellular assays. Normalized concentration-response curves were fitted using a four-parameter logistic equation performed using software developed internally (Helios). The equation

used is for a one-site sigmoidal dose response curve: $Y = A + ((B-A)/(1 + ((X/C)^D)))$, where $A$ = min, $B$ = max, $C = IC_{50}$, $D$ = slope. By default, min is fixed at 0, whereas max is not fixed.

When drugs had no significant biological activity at the highest concentration tested, the $AC_{50}$ was reported with qualifier >; for example, an $AC_{50}$ is reported with qualifier > and $AC_{50}$ value 30 when a compound exhibits no significant activity at concentrations up to 30 μM. Where curve fitting produces an $AC_{50}$ value below the highest concentration tested, activity is reported with qualifier =.

### Mapping chemicals to DrugCentral structure IDs
Multiple chemical substances were tested in the SPD assays. Different substances include distinct lots sourced from chemical vendors or synthesized internally of a given drug, different salt forms, and/or metabolites of the parent drug. In preparation of the dataset released with this study, SMILES representation of substances were de-salted and converted to InChI keys using RDKit's MolToInchi function (https://rdkit.org/; accessed 09/22/2021; RDKit version 2021_03_5).

The DrugCentral PostgreSQL database dated 09/18/2020 was downloaded (https://drugcentral.org/download; accessed 09/22/2021). The InChI key consists of three parts separated by hyphens, of 14, 10, and 1 character(s), respectively. These correspond to the connectivity information (or graph; 14 characters), remaining layers (10 characters), and protonation state (1 character). For each NIBR structure, matches to DrugCentral were attempted at multiple levels of decreasing stringency: (1) perfect matches: the InChI key obtained on the DrugCentral SMILES matches the key from a NIBR SMILES, (2) match without the protonation part of the key, (3) match using only the graph part of the key, but require a name or synonym match, and finally (4) try to match on name or synonym, and review structures. Level 4 matches are common with complex drugs such as natural products, where drawing errors occurred during the registration of substances. Matches involving names and/or synonyms compare those from DrugCentral to names assigned as part of the NIBR substance registration. Note that DrugCentral sometimes includes both parent drugs and metabolites, both of which were used in the matching process.

### Dataset summarization
Substances sharing the same InChI key tested in multiple assay runs were summarized into a single numeric $AC_{50}$ for a given InChI vs. assay pair. When one or more $AC_{50}$ values had qualifier =, the geometric mean was computed and reported with qualifier =; N summarized indicates the number of averaged $AC_{50}$ values, and N total indicates the total number of $AC_{50}$ values for the InChI vs. assay pair, including $AC_{50}$ values with qualifier > excluded from the geometric mean computation. In the absence of any $AC_{50}$ value with qualifier =, the largest value among those with qualifier > was retained. For instance, the $AC_{50}$ values of >1 and >30 μM are summarized as follows: qualifier >, numeric $AC_{50}$ value 30, N summarized 2, and N total 2.

### Defining assay groups
The SPD database was created over several years, with some targets having multiple assay protocols employed. Each protocol was assigned a unique identifier. Changes in protocol often result in the creation of a new assay, designated with a new identifier, even when both assays effectively measure the same biochemical event. Examples include changes in radioligand, measurement technology (e.g., filter binding vs. TR-FRET), outsourcing to a contract research organization (CRO), etc. To maximize power for detecting statistically significant relationships and simplify the analysis, we defined assay groups that combine assays resulting in concordant $AC_{50}$ values when comparing results on the same drug substances.

Concordance analysis was performed by evaluating the agreement of assay results where at least ten compounds were tested in each

pair of assays having the same target and mode (e.g., both binding, agonist, or antagonist assays). Qualitative agreement (<10 and ≥10 μM) was assessed by calculating the sensitivity of each assay for detecting actives from the other, with a minimum of 0.5 for both assays (i.e., # active in both assays / # active in assay 1 ≥ 0.5 and # active in both assays / # active in assay 2 ≥ 0.5, and Pearson $R \geq 0.7$ calculated on 10 or more log $AC_{50}$ values, where both results had qualifier =. Assay pairs with insufficient overlapping test compounds were not merged. Viewing assays as nodes and concordant assay pairs as edges, all nodes within a connected graph were merged, even if some assay pairs within the group fell below our concordance cutoff (or lacked sufficient overlapping pairs to assess). To increase the number of overlapping test compounds for a pair of assays, this analysis was performed on all results available for the assays, including proprietary compounds not included in the supplement. Supplementary Data 1 is provided at the level of both individual and grouped assays.

Within each assay group, the assay supplying the largest proportion of results was designated as the preferred assay. When results were available for the preferred vs. other group assays, results from the preferred assay were used in downstream analyses.

## Integration of external activity results

Activity data from DrugCentral, including annotation of targets as drugs' mechanism of action (MOA), were obtained from the act_data_full table for humans and mammals (rat, mouse, cow, guinea pig, rabbit, pig, sheep, dog, chicken, and monkey). Drugs from the SPD were mapped using the DrugCentral structure ID as described above. For each DrugCentral target ID, the Swissprot identifier from the target_component was mapped to Entrez gene IDs using the gene2accession file from Entrez gene or the Uniprot ID mapping tool (https://www.uniprot.org/uploadlists/; accessed 09/23/2021) in the absence of an Entrez match. Finally, any non-human Entrez gene IDs were mapped to the human ortholog using a compilation of associations from Ensembl, Homologene, RGD, and MGI. For SPD, assays using non-human proteins were represented with the human Entrez gene ID. Each drug vs. assay pair from SPD was annotated with the median $AC_{50}$ for all DrugCentral activity records with the same structure ID and human gene ID. This matching did not consider activity mode (inhibitor, antagonist, etc. - action_type in act_data_full), because it was undefined in most cases.

When defining on-target activity, the DrugCentral act_type variable defining a drug's mode of action at the target was compared to the assay mode. For functional assays, only drugs annotated as having the same mode as the assay were retained (e.g., agonist drugs for agonist assays). For GPCR and nuclear receptor assays having binding or inhibition modes, agonist drugs were removed. Antagonist drugs were retained because binding and functional antagonist readouts are correlated on our panels.

ChEMBL version 27 was downloaded as a PostgreSQL database. To increase the number of matches while allowing variation in structural representations (e.g., ignoring chirality, structure drawing errors, etc.), ChEMBL compound identifiers provided by DrugCentral were supplemented with those matching the SPD InChi keys identified using the multi-criteria match described above. ChEMBL targets identified with Uniprot accession numbers were mapped to SPD assays as described above. Activity types and units suggestive of multiple concentration testing were converted to $AC_{50}$ values in μM units (pM, nM, μM, mM, and M); results with units of ng/mL, pg/mL, and ug/mL were converted to molarities using the drugfree base RDKit molecular weight. ChEMBL results suggestive of single concentration testing (units of %) were considered separately. Multiple ChEMBL drug vs. target values were summarized as the median, separately for $AC_{50}$ and single concentration results.

Clarivate Cortellis (https://clarivate.com/cortellis/solutions/preclinical-intelligence-analytics/; accessed 8/3/2021) and Excelra

GOSTAR (https://gostardb.com/gostar/; accessed 8/3/2021) are subscription-based resources similar to ChEMBL and DrugCentral. The same processing applied to ChEMBL was used to identify median $AC_{50}$ values for SPD drug vs. assay pairs.

## Selecting representative SPD result for each drug vs. assay group pair

Multiple InChi keys are available for certain drugs, and multiple assays within assay groups. To simplify the analysis, we denoted one result as a representative among all available for a given prescribed (parent) drug vs. assay group pair. For a given drug vs. assay group pair, the algorithm selects assay results for the active metabolite over any collected for the parent drug (e.g., take activity results for enalaprilat when describing the activity of enalapril). The algorithm favors high quality structural matches between DrugCentral and SPD and selects from the preferred assay within the assay group. Detailed logic is available in the Jupyter notebook make_all_prescribable_drug_activity_dataset.ipynb. The selected activity records have column representative_result_drug_assay_group = TRUE in Supplementary Data 1.

## Comparison of activity results to ChEMBL

To model assay and target characteristics that may affect concordance between SPD and literature-reported activity results, we assembled a dataset matching SPD activity results with each individual ChEMBL activity reported for a given drug and target pair (i.e., not averaging results for a given drug and target across publications). Only pairs where the activity qualifier was = in both sources were retained. SPD assay results were annotated with assay characteristics reported in Supplementary Data 3, and ChEMBL results with the ChEMBL assay type (e.g., B for binding or F for functional), assay format using the Bioassay Ontology terminology, and standard_type variables from the ChEMBL activity database table. Results were excluded when they were represented by fewer than 500 pairs in the dataset of ca 22 000 SPD vs. ChEMBL result pairs: ABCB11 (SPD-Event = incorporation assay), SCN5A and CACNA1C (SPD Readout = electroanalytical readout), ACHE (SPD Readout = absorbance readout), SPD Protein Class = Protease, ChEMBL bao_format of organism, mitochondrion, microsome, cell-free (blood assays), ChEMBL standard_type not one of Ki, Kd, $IC_{50}$, EC50, ChEMBL assay_type of T or A. Several annotations were merged: SPD Mode of Binding and inhibition, ChEMBL bao_format = subcellular format (mostly brain synaptosomes) with tissue-based format. The final dataset consisted of 21 596 matched SPD vs. individual ChEMBL activity results involving the same drug substance and target (including ortholog matches across species). Activities were converted to $pAC_{50}$ values (negative log10 of $AC_{50}$ in molar units), and the absolute difference was calculated. The activity difference was modeled using continuous variables of SPD $pAC_{50}$, ChEMBL $pAC_{50}$, a binary variable same_species (1 = yes, 0 = no), and multi-level categorical variables SPD Protein Class (e.g., GPCR), SPD-Event (e.g., protein binding assay), SPD Format (e.g., in vitro assay with cellular components), SPD Mode (e.g., Binding), SPD Readout (e.g., radioactivity), ChEMBL assay_type (e.g., = B for binding assays), ChEMBL bao_format (e.g., assay format), ChEMBL standard_type (e.g., IC50). All categorical variables were converted to binary dummy variables with the R library fastDummies. A penalized lasso regression model was fit using the R library glmnet to identify variables that were associated with activity differences. Variables discussed in the text were present in models up to and including the 1se model (model with the fewest variables within 1 standard error of the best model) and had an absolute coefficient greater or equal to 0.1. The modeled dataset is provided in Supplementary Data 10.

## Human drug exposure (Cmax) and plasma protein binding

The maximal drug exposure (Cmax) at the highest approved dosage was curated from the primary literature for 487 drugs. To broaden the

  

dataset, we extracted human Cmax values from Pharmapendium (https://pharmapendium.com; accessed 10/26/2021). These are often from heterogeneous sources, and include results from lower doses, metabolites, pediatric studies, Cmax at steady state, etc. Selecting a representative value could be achieved by calculating the median, average, 3rd quartile, or 90th percentile. We employed the 487 manually curated results as a reference set and found maximal correlation when Pharmapendium values were summarized as the 3rd quartile. This supplied a further 451 drug Cmax results. Finally, we used values reported in ref. 15 for a further 146 drugs, yielding a dataset of 1084 marketed drug Cmax values (Supplementary Data 2).

To calculate free Cmax values, we employed a similar approach for compiling plasma protein binding (PPB) %: primary literature curation (572 drugs), Pharmapendium (332), DrugCentral (90), and Smit et al. (52), providing a dataset of 1045 plasma protein binding. For Pharmapendium, results reported as albumin or glycoprotein binding were excluded because correlation vs. our curated dataset was low; the 3rd quartile provided the highest concordance vs. curation, as observed for Cmax.

Cmax(free) was determined as the product $(100\text{-PPB\%})\times \text{Cmax(tot)}$ for 940 drugs having both parameters available.

## ADR annotation using FAERS and SIDER

Annotation of drugs as being positive or negative for MedDRA-coded ADRs was performed using two sources. In the absence of freely available high-quality manual curation of ADRs and their frequency from labels for all FDA-approved drugs[64], surrogate approaches must be employed. The FDA adverse drug reaction reporting system (FAERS) is often used in pharmacovigilance research to detect ADRs. In this work, we used FAERS data from DrugCentral[28] without any further post-processing. ADRs with a likelihood ratio test (LRT) ≥5 times the drug-specific threshold value were deemed positive for a given drug[14]; otherwise, the drug was labeled as negative for the ADR. Results obtained using an LRT cutoff of 2 are broadly similar (Supplementary Notes).

The SIDER database provides annotation of drug ADRs obtained from text mining applied to drug labels[16]. SIDER uses drug annotation from STITCH, a sister database (http://stitch.embl.de/download/chemicals.v5.0.tsv.gz; accessed 09/23/2021). SMILES from STICH were converted to InChi keys and matched to DrugCentral structures using the multi-criteria matched described above. The mapping is provided in Supplementary Data 11.

Drug ADRs were obtained from the file meddra_all_se.tsv.gz available from the SIDER website (http://sideeffects.embl.de; accessed 09/24/2021). Drugs were mapped to DrugCentral using the stereo_id using Supplementary Data 11. SIDER ADRs were labeled using UMLS CUIs; they were mapped to MedDRA preferred term (PT) codes using the UMLS REST API (https://documentation.uts.nlm.nih.gov/rest/home.html; accessed 09/24/2021). A version of meddra_all_se.tsv, using DrugCentral struct_id and MedDRA PT codes is provided as a supplementary dataset with the Jupyter notebooks (final_sider_map_to_drugcentral_meddra.txt).

## Creating ADR training sets

For each ADR, positive drugs (causing the ADR) and negative drugs (not causing the ADR) must be defined. To study the association between ADRs and assays, ADR terms from SIDER and FAERS reported as MedDRA preferred terms (PTs) were mapped to MedDRA high-level terms (HT) and high-level group terms (HG). The mappings were obtained using the UMLS API. Mapping to higher terms results in more drugs labeled as positive (i.e., higher power to detect an effect), but potentially combining PTs with distinct target (assay) risk factors. We therefore modeled relationships at three levels: PT, HT, and HG.

Depending on the level of MedDRA terms, different strategies were employed for defining ADR-negative drugs (i.e., drugs that do not

cause a given ADR). For HGs, any drug not positive for one of the underlying PT terms was considered a negative. For HTs, any drug not positive for the term *and* not positive for a sister HT under the current HG was considered a negative. For PTs, any drug not positive for the term *and* not positive for a sister PT under the current HT was considered negative. This strategy was used to avoid labeling as negative a drug that produces a similar ADR to the one under study. For example, negatives for PT 0044066 (Torsade de pointes) would exclude drugs that are positive for PT 10047302 (Ventricular tachycardia), because both PTs share the HT 10047283 (Ventricular arrhythmias and cardiac arrest). The Jupyter notebook make_ADR_training_sets.ipynb was used to automate the definition of positive and negative drugs for each PT, HT, and HG term.

Because drugs sharing an active metabolite may have different ADR annotations (e.g., betamethasone dipropionate vs. betamethasone valerate), each was treated separately. Annotation of drugs with a selected activity record from SPD for each assay was performed using the multi-criteria approach described above, using the Jupyter notebook make_ADR_vs_activity_dataset.ipynb. It should be noted that this differs from make_all_prescribable_drug_activity_dataset.ipynb, where we selected a single representative drug form (betamethasone) among all those tested.

## Univariate ADR vs. assay association

To establish the strength of association between drugs' status for ADRs (Boolean) and assay activity measure (continuous measures of $AC_{50}$, free and total margin), the Kruskal–Wallis (KW) test and ROC AUC computation was performed for each ADR vs. assay pair; KW $p$ values were not adjusted for multiple hypothesis testing. ROC AUC values were calculated with the sklearn.metrics.roc_auc_score function using each of the three activity measures as predicted values, and the ADR class (positive = 2, negative = 1) as actual values.

Activity measures of $AC_{50}$, total margin, and free margin frequently have qualifier >, indicating that measured $AC_{50}$ was estimated to exceed the highest concentration tested in the assay. The maximum tested concentrations of 10 and 30 μM were employed for most assays. To calculate a rank-based association test between assays and ADRs, it was necessary to select an $AC_{50}$ cutoff and replace all values in excess with the cutoff value (truncating). Values with qualifier > but $AC_{50}$ below the cutoff were excluded. For $AC_{50}$ values, the numeric distribution for qualifier = and > were largely non-overlapping, the natural cutoff is 10 or 30 μM depending on the assay, and few values needed to be truncated or excluded. Because drug total and free Cmax vary over a wide range, safety margin distributions overlap significantly for qualifier = and >. This makes the selection of cutoff more difficult: too low and one loses the ability to distinguish ranks for drugs with a safety margin above the cutoff, but too high and one must exclude from analysis many values with qualifier > below the threshold (and hence a loss of power). We performed tests using cutoffs of 10 and 30 μM for $AC_{50}$, 2 and 10 for total margin, and 10 and 100 for free margin.

For assessing the statistical significance of literature-reported assay vs. ADR associations (see below; Supplementary Data 12), we retained the threshold giving the smallest (most significant) KW $p$ value. For systematic analysis of all assay vs. ADR pairs, we selected a single threshold per assay in order to minimize the number of tests performed (i.e., increasing the false discovery rate). For each assay and activity measure ($AC_{50}$, total margin, free margin), the cutoff providing the largest total number of assay vs. ADR associations with ROC AUC ≥0.7 and KW $p$ value ≤1e-06 was selected and used for all further analyses. The higher cutoffs (30/10/100 for $AC_{50}$/total margin/free margin) were generally selected (37 vs. 23 assays for $AC_{50}$, 28 vs. 14 assays for total margin, and 35 vs. 5 assays for free margin cutoff).

Excluded from analysis were all combinations of assays, activity measures ($AC_{50}$, free margin, total margin), and ADRs not meeting the following criteria: ten or more positive drugs (i.e., drugs with the ADR),

  

50 or more negative drugs and ten or more non-qualified activity values. Univariate analyses were performed using the Jupyter notebook calc_ADR_vs_assay_score.ipynb.

## Multivariate modeling of ADRs

Multiple assays and activity measures ($AC_{50}$, total margin, free margin) may show significant association with a given ADR. Because assay activity is often correlated across related targets (e.g., targets with similar binding pockets), variable selection strategies can be used to select a smaller number of assay and activity measures among all those which met our univariate threshold (KW $p$ value ≤1e-06 and ROC AUC ≥0.7). We used L1-penalized (Lasso) logistic regression to model each ADR outcome (positive or negative) using the subset of assay + activity measures selected by univariate analysis. $AC_{50}$ and margins were log10 transformed prior to modeling.

When modeling ADR outcomes with multiple assays, missing activity values occur when some drugs were not tested in all assays. For each ADR, we required individual assays to reach 70% or greater coverage compared to the assay with maximal drug count; missing values were imputed by using the median.

For each dataset (ADR class as dependent variable and assay activities as independent variables), a sequence of penalties "C" was generated. The average and standard error of ROC AUC at each penalty were determined using 50 trials of leave 20%-out cross-validation. The smallest "C" (or largest penalty) producing a model with ROC AUC within 1 standard error of the best model was selected. This led to the creation of models with few variables, to discern the principal contributors to ADR risk. Variables with zero coefficients have no significant role in explaining the odds of being positive for a given ADR, after accounting for the contributions of variables with non-zero coefficients.

Because small values of $AC_{50}$ or margins indicate activity in an assay, variables with large negative coefficients in the logistic regression model represent assays for which increasing activity results in higher odds of being positive for a given ADR. For coefficients ≥−0.08, the variable was tagged as not in the model. This corresponds to interpreting a 10-fold decrease in $AC_{50}$ or margin being associated with a smaller than 20% increase in odds ratio of observing the ADR. There were occurrences of coefficients >0.1, i.e., indicating decreased risk of the ADR for activity in the assay. These were almost exclusively in models where an expected negative association was present for another activity parameter of the same assay (i.e., $AC_{50}$ had a large negative coefficient and free margin had a small positive coefficient). These were considered excluded from the model (i.e., coefficient of 0 in Supplementary Data 7). Multivariate analyses were performed with the Jupyter notebook build_ADR_vs_assay_model.ipynb.

## Literature-reported target vs. ADR associations

Target-ADR relationships, as published in three key reviews, were obtained from the supplementary material in ref. 15. Because they did not provide the direction of association (target activation vs. inhibition), we reviewed associations from the three publications. Smit et al. mapped terminology from the reviews to MedDRA preferred terms (PT). In reviewing their results, we added some missing associations and refined mappings to MedDRA codes. These are denoted as pre-analysis supplemental terms in Supplementary Data 12.

Because the selection of MedDRA PTs from the literature reviews may differ from their representation in SIDER or FAERS, we examined the frequency of each literature PT code in SIDER and FAERS. Some terms with suspiciously low frequency triggered searches for better terminology. For instance, "Intestinal transit time decreased" is a valid MedDRA PT used in Lynch et al., however, it is not used in SIDER or FAERS. However, both "Gastrointestinal disorder" and "Diarrhea" were identified as substitutes. These additional mappings were added to the Literature vs. MedDRA PT term mapping.

For each combination of MedDRA code and target from the literature, we examined the significance of the association in the SPD. Each association was tested across all SPD assays for the target (median 1, range 1–6), two sources (SIDER, FAERS), three activity measures (total margin, free margin, $AC_{50}$), and two activity cutoffs for denoting active vs. inactive results (total margin 10 vs. 2, free margin 100 vs. 10, $AC_{50}$ 30 vs. 10 μM). As such, 12 to 72 tests (median of 12) were conducted per literature association, and we classified as "marginally significant" those associations with KW $p$ value between 0.05 and 0.001 (nominal $p$ value of 0.001 with 72 tests yields a Bonferroni-corrected $p$ value ~0.07). We only tested associations having at least 10 positive drugs and 50 negative drugs with available assay results; ADRs with counts below these thresholds were typically rare (e.g., death) and were classified as "not tested".

Failure to achieve significance might be due to a poor selection of MedDRA PT for the ADR. For associations classified as marginal, not significant, or not tested, we repeated the statistical testing described above for each PT that shares a given HT with the literature-derived term. For example, Smit et al. mapped "urinary contraction" (given in the Bowes review as an ADR for CHRM3 activation) to the term "Bladder spasm" (MedDRA 10048994). Our dataset only contained six drugs annotated with this ADR (classification "not tested"). However, several MedDRA PTs sharing the same HT met our criteria of KW $p$ value ≤0.001 and ROC AUC ≥0.6. Ordered by increasing $p$ value (most significant first), these include "Urinary retention" (10046555), "Urinary hesitation" (10046542), "Micturition disorder" (10027561), and "Strangury" (10042170). The selection algorithm sorted all related terms by $p$ value, accumulating the number of tests performed, and stopping at first satisfying the above criteria. In Supplementary Data 6, the association of CHRM3 activation with "urinary contraction" (literature ID 364) contains the $p$ value and ROC AUC for "Urinary retention" and is classified as highly significant ($p$ ≤ 1e-06), with distance 1 (i.e., the significant PT and starting PT are connected via 1 intermediate in the network, via the shared HT).

When no PT terms sharing a given HT were identified, terms sharing a high-level group term (HG) were examined. These are encoded as distance 2 in Supplementary Data 6 (i.e., the starting and significant terms are linked via two intermediates: HT then HG. For both the HT and HG expansion, we used a two-step process: first identifying possible related terms, then manually reviewing and confirming them. This is especially important for distance 2 relations where very broad (and sometimes opposite) effects are grouped at the HG level. Only 14 target-ADR pairs were found significant via a shared HG term (distance = 2), but not distance 0 or 1; 11 of these used the term "Ileus paralytic" (10021333) ascribed to "constipation", "gastrointestinal motility decreased", "gastrointestinal transit decreased") reported in the three reviews.

The Jupyter notebook calc_lit_AE_vs_assay_score.ipynb was used to perform this analysis.

The final set of literature-derived target vs. ADR annotations, including those obtained from Smit et al. and our additions via the common HT and HG terms, are provided in Supplementary Data 6. Tabulation, as significant, marginally significant, not significant, or not tested in results, uses the strongest association for any of the individual MedDRA PTs regardless of their source.

Developmental ADRs from Lynch et al.[9] are included in Supplementary Data 6 for completeness but were excluded from result summaries described throughout. Because these are rare effects, only three of these associations meet the criteria of 10 positives and 50 negatives in SIDER and/or FAERS.

## Assay and ADR characteristics vs. validation of target-ADR pairs

Establishing statistical significance of a given target vs. ADR pair requires having a sufficient number of drugs that are positive and

negative for the ADR, and a sufficient number of drugs with measurable activity in the assay. We selected minimal but arbitrary requirements of 10 positives, 50 negatives, and 10 or more non-qualified activity values to test the association. Failure to find significance may simply reflect the limited power of the dataset, i.e., the above cutoffs were not set high enough.

Because the majority of significant literature-reported associations were supported by the $AC_{50}$ activity measure (rather than free or total margin), we repeated the selection process described above to identify the most significant MedDRA code, source (SIDER or FAERS), assay, and activity cutoff for the $AC_{50}$ activity measure only. This avoided combining $AC_{50}$ and margin-based activity measures having different scales, and for which standard percentile values would be non-comparable (free margin = 1 and $AC_{50}$ = 1 μM are not comparable).

Literature-reported target vs. ADR pairs were classified as significant (222 pairs) or non-significant (497 pairs), using the criteria KW $p$ value ≤ 0.001 and ROC AUC ≥ 0.6. Since this analysis used only the $AC_{50}$ activity measure, there are fewer significant associations compared to Supplementary Data 6 (which used $AC_{50}$, free and total margin). To identify families of ADRs more or less likely to be significant, MedDRA PTs were mapped to system organ classes (SOCs), and each literature association was annotated as assigned to (1) or not (0) a given SOC. Some PTs map to multiple SOCs (e.g., "Metabolism and nutrition disorders", "Endocrine disorders"). Several summary statistics that capture the proportion of drugs with potent activity in the assay were selected: percentile values (2.5, 5, 10, 25, or 50th; e.g., assays with many potent drug activities will be represented with smaller percentile values), count of $AC_{50}$ values less than or equal to 100, 500, or 1 μM, and count of drugs with assay results that are positive or negative for an ADR. The dataset used for this modeling is provided in Supplementary Data 13.

Lasso-penalized logistic regression modeling was performed using the R package Glmnet using default parameters [glmnet(x,y,family = "binomial")] and the AUC metric for cross-validation [cv.glmnet(x,y,family = "binomial", type.measure = "auc")].

### Unpublished assay results vs. known drug ADRs
Unpublished SPD activity results, i.e., drug-target pairs not reported in the sources we considered, were compared to known drug ADRs to determine if these unpublished activities might explain the ADRs. The following criteria were applied: (1) the drug has $AC_{50}$ < 5 μM at a given target that is not reported in ChEMBL, DrugCentral, or the subscription resources, (2) the drug is known to cause a given ADR according to SIDER and/or FAERS, (3) literature-reported target vs. ADR relationship is significant in SPD ($p$ ≤ 0.001) on one of more of $AC_{50}$, free or total margin, (4) the drug's activity on one or more of those significant measures is in the top three quartiles among all drugs active in the assay and having the ADR, (5) the drug's on-target activity is not associated with this ADR, (6) any known off-target activities associated with the ADR have significantly lower potency compared to the new activity (≥ 10-fold). The final criteria avoid flagging unpublished weak activities unlikely to make significant additive contributions to the ADR risk.

### Data analysis
IPython version 7.29 was used to run the provided Jupyter notebooks. Other analyses were performed with R 4.2.1. Plots were prepared in R, Excel (Office 365), and TIBCO Spotfire 11.8. Concentration-response curves were produced with GraphPad Prism 9.5.1. Final figures were assembled with Adobe Illustrator 27.6.1.

### Reporting summary
Further information on research design is available in the Nature Portfolio Reporting Summary linked to this article.

## Data availability
A Zenodo repository contains results from the Safety Pharmacology Database (SPD), data files used as inputs for the Jupyter notebooks, including downloads from FAERS and SIDER, and Supplementary Data referenced in the manuscript. The repository is available at https://doi.org/10.5281/zenodo.7378746. Source data for figures and tables are provided in Source Data.xlsx and the supplementary datasets referenced in the text. Source data are provided with this paper.

## Code availability
The associated Python code in the form of six separate Jupyter Notebooks described in the Methods section of the manuscript is available on GitHub (www.github.com/Novartis/SPD).

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

## Acknowledgements

We thank Jonathan Moggs and Greg Friedrichs for their support. We also thank the Translational Medicine Data Science Academy team at NIBR.

## Author contributions

A.F. and LU. provided the secondary pharmacology material, J.J.S., D.Y., and A.F. analyzed and interpreted the data. J.J.S and L.U. drafted the manuscript. All authors critically reviewed the manuscript.

## Competing interests

At the time of preparation of the manuscript, all authors were employees of Novartis Pharma AG. The authors declare no competing interests.
