## [Peer Review File · Nature Communications]

A novel preclinical secondary pharmacology resource illuminates target-adverse drug reaction associations of marketed drugsREVIEWER COMMENTS

Reviewer #1 (Remarks to the Author):

The authors report SPD (Secondary Pharmacology Database) comprising ca 150k AC50 values for 200 drugs, and perform subsequent statistical analyses for target-SIDER and target-FAERS associations. Several (many) novel (in-)activities of compounds on targets, and new target-ADR associations are reported and discussed.

An excellent resource for the community and this referee would certainly recommend publication

A few pointers to what the authors might want to address/discuss in more details:

- I65: "Here, we report overall low concordance between results from the SPD (obtained using a limited number of assay protocols for each target) vs. results from ChEMBL and DrugCentral (obtained using a wide variety of such protocols)."

Yes, but this is plausible, but not yet causal. Might some readout types etc. be 'better' (translatable, say functional vs binding assays, etc., as well reproducible) than others? Probably the authors gained quite a lot of insight over the years, so this could be discussed in more depth (I have seen section Defining assay groups in methods and associated Dataset S1, but this could be more prominent, since of interest to the reader, also in the results section - 'so which type of assay should we now run in practice?')

- Text in Figure 2 B is difficult to read

- I140: "Drugs with higher overall promiscuity, defined as the percentage of assay groups with AC50 \leq 10 μ M, contributed a significant portion of known and novel physiological off-target activities (Fig. 2B)"

I don't think the figure supports this (global) statement directly, since it only plots data for drugs with high overall promiscuity. It's plausible, but numbers should be provided that support this precise statement

- I 146-162 please make sure statements in the text (e.g. about clinical use) are fully supported by references

- I170: "Evaluation of literature-reported target – adverse drug reaction relationships"

How were frequency, severity, reporting biases etc taken into account? This gets discussed to an extent later, but in the experience of the referee this (ADR data) is the most difficult type of chemical and biological data to deal with due to so many reasons, this should be appreciated also in the text appropriately

- Figure 3: in this context a precision/recall analysis could be useful, often there is a significant trade-off between target-ADR associations here

- I327-331 please make sure statements are fully supported by references

- I472-479 - I think the limitations of FAERS and SIDER could be discussed in more detail (various biases, such as reporting etc.; also how frequency, severity etc. are dealt with, this is really tricky with adverse event data). E.g. l. 663, is the choice of LLR \geq 5 having an impact on the subsequent results? Could/probably should be discussed more in discussion. Many choices (statistics, which subsets of data not to use due to small size etc.) are subjective, but e.g. binarization choices can have a real impact on results, hence ideally investigate sensitivity to key parameters, but please at least add discussion to respective section

That being said, a very valuable addition to the field, both w.r.t. data made available and the analyses performed, and this referee would recommend publication of the work ideally after the above comments have been addressed

Reviewer #2 (Remarks to the Author):

The authors construct, analyze, and release an impactful new resource for adverse drug reaction (ADR) prediction, the Secondary Pharmacology Database (SPD). Despite longstanding efforts, the off-target binding activities of drugs and drug candidates are incomplete, even for protein targets with known ADR associations. The SPD's ~150,000 new drug-target data points contribute years of otherwise unreported observations to a field whose public datasets too frequently suffer from data sparsity, narrow scopes, or inconsistent assay formats. Several notable results arise from the analysis, from the comparative novelty of SPD's negative results (95% vs. other pharmacology databases) to the meaningful disagreement with 64% of putative ADR-target links reported in the literature. The authors acknowledge the unexpected observation that unadjusted drug-target AC50s were no less effective in logistic regression models predicting ADRs from off-target binding than margin-based features (which may achieve higher precision, however). In short, the SPD and the author's analyses comparing it to drug-target and, subsequently, the target-ADR associations derived from public databases such as ChEMBL, FAERS, and SIDER are intriguing, and the release of SPD itself will likely prove a boon to the ADR field and as an actionable resource for subsequent researchers.

MINOR POINTS

1. Line 344: Selection of informative targets via multivariate logistic regression with lasso penalization. Per the discussion on disentangling potentially equally-informative targets, e.g., DRD3 vs. DRD2 for motor dysfunction, the model may break ties in feature selection stochastically during multivariate analysis, depending on the implementation. If so, random-seeded repeats of model training would show an asymptotically equal selection of interchangeable targets across the repeats. Could comparing feature selection across repeated model training distinguish groups of interchangeable vs. stably-selected targets?
2. The citalopram-HRH2-GERD/GI logic seems flipped unless I'm misreading. E.g., depression & tricyclic antidepressant use lead to increased GERD; HRH2 inhibition combats GERD; therefore, citalopram's newly observed HRH2 inhibition would combat or at least no longer increase GERD. Or could this be clarified concerning the mixed reports of GERD with citalopram use?
3. Fig 2B: The y-axis is difficult to read.
4. Line 181: Can the authors confirm this is not the Kruskal-Wallis test (a non-parametric alternative to the one way ANOVA)? There are multiple spellings online if we are thinking of the same test.
5. Line 184: How is the ROC calculated? I imagine it is on the predictions from the univariate logistic regression models introduced later in the text. Or was this ROC calculated over some other range of scoring thresholds instead?

Response to reviewer comments for

A novel preclinical secondary pharmacology resource illuminates target-adverse drug reaction associations of marketed drugs

We have answered each critique of the Reviewers, wherever possible, simply by adopting their suggestions. We are grateful for the attention they clearly lavished on the manuscript. Both have strengthened the manuscript

Reviewer questions in bold

Our responses in red

Additions to the text in red italics

Reviewer #1:

- I65: "Here, we report overall low concordance between results from the SPD (obtained using a limited number of assay protocols for each target) vs. results from ChEMBL and DrugCentral (obtained using a wide variety of such protocols)."

Yes, but this is plausible, but not yet causal. Might some readout types etc. be 'better' (translatable, say functional vs binding assays, etc., as well reproducible) than others? Probably the authors gained quite a lot of insight over the years, so this could be discussed in more depth (I have seen section Defining assay groups in methods and associated Dataset S1, but this could be more prominent, since of interest to the reader, also in the results section - 'so which type of assay should we now run in practice?)

We acknowledge that this is a fair point and we recognized how difficult it is to investigate systematic differences in ChEMBL vs our internal assays, owing to very limited annotation of the ChEMBL assays using terminology from Bioassay Ontology (BAO). For instance, of ~13,000 ChEMBL assays that align with the ~100 assay groups in our database, only 4 have annotation in the assay_class_map and 187 have detailed assay conditions in the assay_parameters table of the ChEMBL schema. We therefore limited our investigation to the role of annotation of the SPD assays (Table S3), and limited ChEMBL assay attributes as noted below:

In results, on page 5, bottom paragraph, we added:

To systematically investigate factors contributing to activity differences, we matched SPD vs. 21 596 individual ChEMBL activity results, and annotated each activity pair using assay and target attributes (methods). When modelling differences in log AC₅₀ values, SPD attributes denoting agonist assays (“Mode”), kinase assays (“Protein Class”), and protein functional assays (“Event”, e.g. calcium flux assays as opposed to binding assays) tended to increase differences, as did ChEMBL attributes “protein format” (a Bioassay Ontology assay type typically denoting assays using radioligand displacement in brain homogenates) or ChEMBL standard type of “EC₅₀” (typically associated with functional assays). SPD attributes denoting binding assays (Mode = “Binding” or “inhibition”) were associated with smaller activity differences. As noted above, activity differences tended to be larger when the reported activity was higher in ChEMBL. Notably, comparing assays across species (e.g. human vs. mouse protein) was not associated with larger activity differences. Taken together, these represent “received wisdom” in comparing assays across sources: assays measuring functional events downstream of targets are more variable than those measuring binding events at targets. These trends are likely to be confounded by association of measurement approaches and target families difficult to distinguish in our database (e.g. kinase/enzyme assays are cell-free assays and GPCRs or ion channels are cell-based assays).

In methods, on page 27 bottom paragraph, we added:

To model assay and target characteristics that may affect concordance between SPD and literature-reported activity results, we assembled a dataset matching SPD activity results with each individual ChEMBL activity reported for a given drug and target pair (i.e., not averaging results for a given drug and target across publications). Only pairs where the activity qualifier was “=” in both sources were retained. SPD assay results were annotated with assay characteristics reported in Supplementary Data 3, and ChEMBL results with the ChEMBL assay type (e.g. “B” for binding or “F” for functional), assay format using the Bioassay Ontology terminology, and standard_type variables from the ChEMBL “activity” database table. Results were excluded when they were represented by fewer than 500 pairs in the dataset of ca 22 000 SPD vs. ChEMBL result pairs: ABCB11 (SPD Event = “incorporation assay”), SCN5A and CACNA1C (SPD Readout = “electroanalytical readout”), ACHE (SPD Readout = “absorbance readout”), SPD Protein Class = “Protease”, ChEMBL bao_format of “organism”, “mitochondrion”, “microsome”, “cell-free” (blood assays), ChEMBL standard_type not one of Ki, Kd, IC50, EC50, ChEMBL assay_type of “T” or “A”.

Several annotations were merged: SPD Mode of "Binding" and "inhibition", ChEMBL bao_format = "subcellular format" (mostly brain synaptosomes) with "tissue-based format". The final dataset consisted of 21 596 matched SPD vs. individual ChEMBL activity result involving the same drug substance and target (including ortholog matches across species). Activities were converted to pAC50 values (negative log10 of AC50 in molar units), and the absolute difference calculated. The activity difference was modelled using continuous variables of SPD pAC50, ChEMBL pAC50, a binary variable same_species (1="yes", 0="no"), and multi-level categorical variables SPD Protein Class (e.g. "GPCR"), SPD-Event (e.g. "protein binding assay"), SPD Format (e.g. "in vitro assay with cellular components"), SPD Mode (e.g. "Binding"), SPD Readout (e.g. "radioactivity"), ChEMBL assay_type (e.g. = "B" for binding assays), ChEMBL bao_format (e.g. "assay format"), ChEMBL standard_type (e.g. "IC50"). All categorical variables were converted to binary dummy variables with the R library "fastDummies". A penalized lasso regression model was fit using the R library "glmnet" to identify variables that were associated with activity differences. Variables discussed in the text were present in models up to and including the "1se" model (model with fewest variables within 1 standard error of the best model) and had absolute coefficient greater or equal to 0.1. The modelled dataset is provided in Supplementary Data 10.

- Text in Figure 2 B is difficult to read

The figure has been remade, per next point and reviewer 2

- I140: "Drugs with higher overall promiscuity, defined as the percentage of assay groups with $AC_{50} \leq 10 \mu M$, contributed a significant portion of known and novel physiological off-target activities (Fig. 2B)"

I don't think the figure supports this (global) statement directly, since it only plots data for drugs with high overall promiscuity. It's plausible, but numbers should be provided that support this precise statement

We changed the format of the figures 2 and 3 to address this point and included all 894 drugs having 30 or more assay results (i.e., sufficient assay data to estimate promiscuity) and free Cmax available. We believe that the new version is clearer. We added a supplementary figure to make the point another way, and provided the following sentence in results:

On page 7, 2nd paragraph: *For instance, the promiscuous antidepressant nefazodone (31/88 assays with AC50 results < 10 µM, or 35%) has 4 on-target and 23 off-target physiological activities; 5 off-target activities were not reported in the sources we considered. There are outliers from the overall trend: sunitinib has 51% target promiscuity, yet only a single physiological activity (on-target), owing to its very low free C_{max} of 6.3 nM; the antibiotic cefepime has no AC50 results < 10 µM, yet 6 off-target activities above this threshold may be physiologically relevant owing to its high free C_{max} of 260 µM (Supplementary Fig. 1).*

- I 146-162 please make sure statements in the text (e.g., about clinical use) are fully supported by references

We inserted the required references 18, 23 and 24 accordingly starting on page 7, bottom paragraph: *(the CNR1 antagonist rimonabant is used in the management of obesity¹⁸) ... Depression²² and the use of tricyclic antidepressants have been reported to increase the incidence of gastro-esophageal reflux disease (GERD), but not SSRI antidepressants²³ (of which citalopram is the most prescribed²⁴).*

- I170: "Evaluation of literature-reported target ØC adverse drug reaction relationships"
How were frequency, severity, reporting biases etc taken into account? This gets discussed to an extent later, but in the experience of the referee this (ADR data) is the most difficult type of chemical and biological data to deal with due to so many reasons, this should be appreciated also in the text appropriately

We agree with the reviewer that both SIDER and FAERS have significant limitations, but they are used in our work and many others in the field because there are no publicly available resources of human curated ADRs from all drug labels, including severity and frequency. We elaborated on these points in discussion starting on page 20, 1st paragraph:

Methods used in this work to annotate drugs with ADRs have limitations. Expert annotation of structured product labels is limited to small drug sets⁶⁴, and revealed limitations of natural language processing (SIDER) or post-marketing spontaneous reports as approximations. Severity and frequency are not generally available in SIDER, and hence not reflected in the associations described herein. Inferring ADRs from FAERS has several pitfalls, including reporting biases and confounding by drug indication¹². Various statistical approaches have been proposed to extract

trends from FAERS data⁶⁵⁻⁶⁷, and consensus on the most effective approach remains elusive. Our work leveraged the Likelihood Ratio Test⁶⁸ as implemented in DrugCentral; other approaches may yield different results.

- Figure 3: in this context a precision/recall analysis could be useful, often there is a significant trade-off between target-ADR associations here

We agree that precision (also known as positive predictive value or PPV) is an important quantity to convey when using predictions about adverse drug reactions (or other toxicities), especially when the prevalence of positives in the application domain is low. This is common in toxicity prediction for clinical candidates, where overtly toxic compounds have been eliminated earlier in the drug discovery process. What complicates the analysis is that prevalence of positives in the training set (e.g. the percent of drugs associated with long QT syndrome among all drugs, including drugs first marketed decades ago) may not be the same as the application domain (clinical candidates for newly pursued targets). Further, the prevalence in the application domain is often unknown (e.g. prevalence of long QT in recent drug candidates ... is it the recent industry average, or a higher / lower rate for a new modality like targeted protein degradation?). It's therefore necessary to show users a range of PPV values at different possible prevalence values). Predictors with very high sensitivity and specificity (and hence ROC AUC) can have low PPV owing to low prevalence.

In Figure 3, we focused on assessing whether target-ADR relationships are statistically significant using permissive criteria of p-value < 0.001 from a KW test, and a minimal threshold ROC AUC \geq 0.6. The reviewer's question refers to choices made to operate at different points along a ROC AUC curve, i.e. increasing the score cutoff used to delineate positive vs negative predictions, and therefore increasing specificity at the expense of sensitivity. We therefore added results using a partial ROC AUC over the 90-100% specificity interval, and show high correlation with both the (log) KW p-value and ROC AUC (Supplementary Notes).

On page 8, 1st paragraph (results), we added "*Similar results were obtained using alternate FAERS risk or assay score thresholds (Supplementary Notes)*".

In Supplementary Notes, we added the section "Investigation of assay activity thresholds for defining assay positives vs. negatives":

Observing a statistically significant relationship for a given target and ADR does not provide guidance on its use for assessing compounds. For example, the relationship between hERG (KCNH2) and “Prolongation of QT interval” (MedDRA 10014387) has KW-p-value 1.6e-13 and ROC AUC 0.68 (assessed using SIDER; Supplementary Data 6). Our preferred approach is to report the odds of observing the ADR in clinical use given the measured level of activity; there is no benefit in declaring compounds as being “active” or “inactive” in the hERG assay by thresholding on the free margin or AC50. When thresholding is applied, there is a trade-off between sensitivity (identifying all the QT prolonging drugs) and specificity (falsely labelling a safe drug as QT prolonging). The threshold may change during drug discovery, with a preference for avoiding false positives in later stages. As such, the partial ROC AUC assessed at high specificity may be preferred over the full AUC. To investigate the impact of this threshold, we calculated the partial ROC AUC over the 90-100% specificity interval, and compared to the full (standard) ROC AUC and KW p-value over the full dataset of assay vs ADR pairs (Supplementary Fig. 6; dataset produced by the `calc_AE_vs_assay_score.ipynb` notebook). The high correlation observed indicates that results using this partial AUC would be broadly comparable to those using the standard AUC. We favor the standard AUC because of its familiarity and simple interpretation: the ROC AUC conveys the probability that a randomly selected drug positive for an ADR is ranked above a randomly selected negative drug. A ROC AUC of 0.5 indicates a random result. Partial AUCs are smaller because they measure a fraction of the full specificity interval, and there is no single standard cutoff like 0.5 that corresponds to random accuracy.

We also implemented area under the precision-recall curve (AUPRC; from scikit learn `sklearn.metrics.average_precision_score`), a metric we have not used prior to this work. AUPRC was not correlated with ROC AUC or (log) KW-p-value. Absence of correlation with a standard statistical test is counter intuitive. However, AUCPR vs the prevalence of positive drugs had $R^2 > 0.96$ across 3 activity measures (AC50, free margin, total margin) and 2 sources (SIDER, FAERS). We suggest that AUCPR has lower utility in the context of this work (association of targets and ADRs), and reflects well-known challenges in applying predictive models in low prevalence environments.

- I327-331 please make sure statements are fully supported by references

Thanks to the Referee for pointing out this overlooked matter. Now we have provided the relevant references 29 and 30 for our statements starting on page 13, 2nd paragraph: *“Amongst these, the glucocorticoid receptor ...”*.

- I472-479 - I think the limitations of FAERS and SIDER could be discussed in more detail (various biases, such as reporting etc.; also how frequency, severity etc. are dealt with, this is really tricky with adverse event data). E.g. I. 663, is the choice of LLR \geq 5 having an impact on the subsequent results? Could/probably should be discussed more in discussion. Many choices (statistics, which subsets of data not to use due to small size etc.) are subjective, but e.g. binarization choices can have a real impact on results, hence ideally investigate sensitivity to key parameters, but please at least add discussion to respective section

We refer to discussion of FAERS / SIDER limitations above. To address the point about LRT threshold (changed from less accurate acronym LLR in the original submission), we repeated analyses of the literature-reported target-ADR pairs using a LRT cutoff of 2.

On page 29, 1st paragraph (methods), we added: *“Results obtained using a LRT threshold of 2 are broadly similar (Supplementary Notes)”*

In Supplementary Notes, we added a section subtitled *“Investigation of FAERS likelihood ratio test (LRT) threshold”*

Drug vs. ADR risk from FAERS are annotated with a LRT statistic in DrugCentral. Increasing the LRT threshold for distinguishing drugs annotated as positive (above the LRT) for a given ADR may focus on the smaller set of drugs with higher incidence of the ADR, at the expense of reducing the count of drugs annotated as positive and hence power to detect a significant relationship. The selection of a threshold is arbitrary, values such as 2, 5 or 10 may be selected. Throughout this work, LRT threshold of 5 was used.

To investigate the impact of using a different threshold, the statistical significance of the literature-reported target-ADR relationships from Supplementary Data 6 was evaluated separately using SIDER, FAERS LRT threshold of 2 and 5. Criteria for significance were the same as Fig. 3, namely KW p-value \leq 0.001 and ROC AUC \geq 0.6. Because we required at least 10 ADR positives with assay results when evaluating each target-ADR relationship, and SIDER vs. FAERS or LRT 2

vs. 5 affects this count, 539 target-ADR pairs with assessed significance on the 3 methods were retained for analysis (Supplementary Data 14). The 3 methods were compared via 2 x 2 contingency tables and χ^2 tests (Supplementary Table 1). Of 114 target-ADR pairs significant at LRT threshold of 5, 86 were also significant at LRT threshold of 2. While all 3 pairs of approaches are highly concordant by the χ^2 statistic, results are more similar when comparing the FAERS LRT thresholds than FAERS vs. SIDER. As such, different LRT thresholds result in broadly similar conclusions.

Reviewer #2 (Remarks to the Author):

1. Line 344: Selection of informative targets via multivariate logistic regression with lasso penalization. Per the discussion on disentangling potentially equally-informative targets, e.g., DRD3 vs. DRD2 for motor dysfunction, the model may break ties in feature selection stochastically during multivariate analysis, depending on the implementation. If so, random-seeded repeats of model training would show an asymptotically equal selection of interchangeable targets across the repeats. Could comparing feature selection across repeated model training distinguish groups of interchangeable vs. stably-selected targets?

In the “build_AE_vs_assay_model” notebook, we run 50 trials for each experiment with different train-test set splitting. We investigated the stability of selected targets as described below.

On page 12, 2nd paragraph (results), we added:

The selection of assays retained in the sparse models was stable across random resampling of the data (Supplementary Notes)

In supplementary notes, we added a section “Investigation of stability of variable selection in multivariate modelling of adverse drug reactions”:

Lasso-penalized logistic regression modelling was used to select non-redundant variables (assay and activity measure) explaining outcomes for each source (SIDER or FAERS) and MedDRA code: 115 ADR models from FAERS and 259 from SIDER. For each model, the optimal value of the shrinkage L1 penalty (parameter “C” in scikit-learn LogisticRegression) was selected by performing 50 trials of leave 20% out cross validation and identifying the most penalized model (smallest “C”) within 1 standard error of the maximal ROC AUC. A single final model was

subsequently created at the optimal parameter using the full dataset, and variables having coefficient ≤ -0.08 in this single model were labelled as non-redundant in explaining the ADR (Supplementary Data 7 column "parameters in sparse model"). It should be noted that Supplementary Data 7 summarizes inclusion of assays using any of the three activity measures: free margin, total margin or unadjusted AC50. Variables as used in the model are a combination of assay and activity measure, e.g. KCNH2 AC50 and KCHN2 free margin are separate variables, only one of which might be selected as non-redundant owing to their correlation. To investigate whether the variables selected as non-redundant would change with variation in the dataset, we compared the coefficient in the single final model to the frequency of that variable's inclusion across the 50 repeats (i.e. selecting variables on the training sets only, inside the cross validation loop). For FAERS, 72% of non-redundant predictors were reselected in 40 or more of the 50 repeats, and 81% for SIDER; 2-4% were re-selected fewer than 25 repeats (Supplementary Table 2). Further, of 221 variables re-selected in fewer than 40 repeats, 63 (29%) involved assays that were re-selected using a different activity measure, e.g. retaining the use of KCNH2 assay, but using free margin instead of total margin (Supplementary Data 15). This indicates that the assays selected as non-redundant predictors of ADR risk, as tabulated in Supplementary Data 6 and elsewhere, are not sensitive to variation in the derivation data.

2. The citalopram-HRH2-GERD/GI logic seems flipped unless I'm misreading. E.g., depression & tricyclic antidepressant use lead to increased GERD; HRH2 inhibition combats GERD; therefore, citalopram's newly observed HRH2 inhibition would combat or at least no longer increase GERD. Or could this be clarified concerning the mixed reports of GERD with citalopram use?

We removed the statement about effects of SSRIs on GERD being mixed, as it seems to confuse the example. The purpose was to clarify that effect is not universally observed as positive, however upon review the negative evidence we cited was from case reports and a small study. The examples here and elsewhere are necessarily speculative – we're pointing out that the novel activities might explain known clinical trends

We clarified as follows on page 8, 1st paragraph, with some added references:

Depression and the use of tricyclic antidepressants increase the incidence of gastro-esophageal reflux disease (GERD), but not SSRI antidepressants²² (of which citalopram is the most

prescribed²³). Clinical studies suggest direct effects (rather than altered pain perception) on esophageal function²⁵.

Fig 2B: The y-axis is difficult to read.

We rectified this matter, please see response to the same question in the answer to Referee 1.

4. Line 181: Can the authors confirm this is not the Kruskal-Wallis test (a non-parametric alternative to the one way ANOVA)? There are multiple spellings online if we are thinking of the same test.

We corrected “Wallace” to “Wallis”. It was a typo. We used the non-parametric Kruskal-Wallis throughout.

5. Line 184: How is the ROC calculated? I imagine it is on the predictions from the univariate logistic regression models introduced later in the text. Or was this ROC calculated over some other range of scoring thresholds instead?

On page 32, 2nd paragraph (methods), we clarified that ROC AUC values in Supplementary Data 6 and 7 are calculated as follows:

ROC AUC values were calculated with the `sklearn.metrics.roc_auc_score` function using each of the 3 activity measures as predicted values, and the ADR class (positive = 2, negative = 1) as actual values.

REVIEWERS' COMMENTS

Reviewer #1 (Remarks to the Author):

All point addressed - a good paper, thanks for this valuable contribution to the field

Reviewer #2 (Remarks to the Author):

The authors have addressed my questions, including performing a stability analysis on their multivariate modeling feature selection. I recommend publication.

Response to reviewer comments for

A novel preclinical secondary pharmacology resource illuminates target-adverse drug reaction associations of marketed drugs

Reviewer questions in bold

Our responses in red

Additions to the text in red italics

Reviewer comments – first revision

Reviewer #1:

All point addressed - a good paper, thanks for this valuable contribution to the field

Reviewer #2:

The authors have addressed my questions, including performing a stability analysis on their multivariate modeling feature selection. I recommend publication.

We thank both reviewers for their comments and analysis, which have helped us improve the manuscript.